EMBO
Molecular Medicine

# HPV/E7 induces chemotherapy-mediated tumor suppression by ceramide-dependent mitophagy

Raquela J Thomas[1,2], Natalia Oleinik[1,2], Shanmugam Panneer Selvam[1,2], Silvia G Vaena[1,2], Mohammed Dany[1,2], Rose N Nganga[1,2], Ryan Depalma[1,2], Kyla D Baron[1,2], Jisun Kim[1,2], Zdzislaw M Szulc[1] & Besim Ogretmen[1,2,*] iD

## Abstract

Human papillomavirus (HPV) infection is linked to improved survival in response to chemo-radiotherapy for patients with oropharynx head and neck squamous cell carcinoma (HNSCC). However, mechanisms involved in increased HNSCC cell death by HPV signaling in response to therapy are largely unknown. Here, using molecular, pharmacologic and genetic tools, we show that HPV early protein 7 (E7) enhances ceramide-mediated lethal mitophagy in response to chemotherapy-induced cellular stress in HPV-positive HNSCC cells by selectively targeting retinoblastoma protein (RB). Inhibition of RB by HPV-E7 relieves E2F5, which then associates with DRP1, providing a scaffolding platform for Drp1 activation and mitochondrial translocation, leading to mitochondrial fission and increased lethal mitophagy. Ectopic expression of a constitutively active mutant RB, which is not inhibited by HPV-E7, attenuated ceramide-dependent mitophagy and cell death in HPV(+) HNSCC cells. Moreover, mutation of E2F5 to prevent Drp1 activation inhibited mitophagy in HPV(+) cells. Activation of Drp1 with E2F5-mimetic peptide for inducing Drp1 mitochondrial localization enhanced ceramide-mediated mitophagy and led to tumor suppression in HPV-negative HNSCC-derived xenograft tumors in response to cisplatin in SCID mice.

**Keywords** ceramide; Drp1; E2F; HPV; mitophagy
**Subject Categories** Cancer; Microbiology, Virology & Host Pathogen Interaction

## Introduction

Human papillomavirus (HPV) infection is linked with several cancers such as cervix and head and neck carcinomas (Killock, 2015; Lehtinen & Dillner, 2013). The potential impact of prophylactic HPV vaccines on the prevention of these cancers is of interest.

However, there are differences in the epidemiology of oral and genital HPV infection, such as differences in age and sex distributions, which suggest that the vaccine efficacy observed in genital cancers may not be directly translatable to HPV(+) head and neck cancers, which are mainly located in the oropharynx (D'Souza et al, 2007; Leemans et al, 2011). Paradoxically, HPV infection is associated with improved survival outcome in response to chemo-radiotherapy for patients with head and neck squamous cell carcinoma (HNSCC), and not in patients with HPV(+) cervical cancers (Fakhry et al, 2008; Ang et al, 2010). However, molecular mechanisms involved in increased HNSCC cell death by HPV signaling in response to therapy are largely unknown.

A novel form of cell death, lethal mitophagy, was recently identified in HPV(−) HNSCC, induced by pro-cell death signaling sphingolipid molecule ceramide (Sentelle et al, 2012). Dynamin-related protein 1 (Drp1) activation is an upstream inducer of ceramide-dependent lethal mitophagy (Sentelle et al, 2012). Upon cellular stress, Drp1 translocates to the outer mitochondrial membrane, where it associates with the mitochondrial fission factor receptor (MFF) and forms oligomers, resulting in mitochondrial fission, leading to targeting of damaged mitochondria by autophagosomes (Smirnova et al, 2001; Strack & Cribbs, 2012). Drp1-mediated mitochondrial fission mediates direct lipid–protein interaction between ceramide accumulated in the outer mitochondrial membrane and microtubule associated protein 1 light chain 3 beta (LC3B), a component of autophagosomes, for selectively targeting/degradation of mitochondria by autophagy (mitophagy) (Dany & Ogretmen, 2015). In some cancer cells, such as HNSCC, targeting mitochondria by mitophagy leads to cell death due to decreased cellular energy and/or reduced mitochondrial metabolism/signaling, altering the production of molecules that are essential for cellular growth including nucleotides, amino acids, and metabolic intermediates (Weinberg & Chandel, 2015).

However, whether induction of ceramide-mediated lethal mitophagy is involved in increased cell death in HPV(+) HNSCC in response to therapeutic treatment and cellular stress has not been described previously. Therefore, we set out experiments to investigate the role and mechanisms by which HPV signaling enhances the

---

1   Department of Biochemistry and Molecular Biology, Medical University of South Carolina, Charleston, SC, USA
2   Hollings Cancer Center, Medical University of South Carolina, Charleston, SC, USA
    *Corresponding author. Tel: +1 843 792 0940; Fax: +1 843 792 2556; E-mail: ogretmen@musc.edu

response of HNSCC versus cervical cancer cells to chemotherapy-mediated cellular stress by ceramide-dependent lethal mitophagy.

# Results

### HPV infection enhances HNSCC tumor suppression and cell death

To determine whether HPV(+) compared to HPV(−) HNSCC cells are more sensitive to stress-mediated growth inhibition in response to chemotherapeutic drugs, we first established HPV(+) UM-SCC-47- and HPV(−) UM-SCC-22A-derived xenograft tumors in SCID mice. After the xenograft-derived tumors were ~75–100 mm$^3$, we treated the mice with cisplatin (3.5 mg/kg at 48-h intervals) for 2 weeks, which is below its maximum tolerated doses (van Moorsel et al, 1999). Then, we measured tumor volumes at days 0 and 14 of treatment. The data suggest that the growth of HPV(+) UM-SCC-47-derived HNSCC xenografts was inhibited in response to cisplatin (~75% inhibition), whereas there was no detectable tumor suppression when HPV(−) UM-SCC-22A-derived xenografts were treated with this drug in SCID mice (Fig EV1A and B). After tumor growth measurements in mice, we surgically removed tumors and measured the effects of cisplatin on CerS1-6 mRNA by qRT–PCR. Cisplatin induced CerS1 and CerS6 mRNA in HPV(+) but not in HPV(−) xenograft-derived tumors (Fig EV1C and D). These data were also consistent when we measured the effects of cisplatin on the growth of these cells in culture, in which HPV(+) UM-SCC-47 cells exhibited ~14-fold sensitivity for cisplatin-induced growth inhibition/cell death compared to HPV(−) UM-SCC-22A cells (Fig EV1E). Similar data were obtained using C$_{18}$-pyridinium-ceramide (C$_{18}$-pyr-cer), a ceramide analogue drug, in which HPV (+) UM-SCC-47 cells exhibited ~20-fold sensitivity for C$_{18}$-pyr-cer-induced growth inhibition/cell death compared to HPV(−) UM-SCC-22A cells in culture (Fig EV1F). The inhibitory concentrations that suppressed growth by 50% (IC50) for cisplatin in UM-SCC-47 [HPV(+)] versus UM-SCC-22A [HPV(−)] cells were 1.6 versus 23 μM, and for C$_{18}$-pyr-cer in UM-SCC-47 [HPV(+)] versus UM-SCC-22A [HPV(−)] cells were 0.6 versus 13 μM at 48 h. Thus, these data suggest that UM-SCC-47 (HPV+) cells exhibit more sensitivity for cisplatin- and C$_{18}$-pyr-cer-mediated growth inhibition and/or cell death than HPV(−) UM-SCC-22A cells/tumors in culture and/or in mice, recapitulating clinical data observed in HPV(+) versus HPV(−) HNSCC patients.

### HPV-E7 signaling induces cell death by CerS1/ceramide-dependent mitophagy

CerS1-generated C$_{18}$-ceramide was shown to induce HNSCC cell death (Venkataraman et al, 2002; Koybasi et al, 2004; Pewzner-Jung et al, 2006). This is confirmed by treatment of mouse embryonic fibroblasts (MEFs) isolated from wild-type (WT) and CerS1-toppler mutant mice (CerS1$^{top/top}$), which encodes for inactive enzyme for defective C$_{18}$-ceramide generation (Zhao et al, 2011; Spassieva et al, 2016) by sodium selenite (SS), a known inducer of mitophagy (Sentelle et al, 2012). SS exposure (10 μM, 3 h) markedly induced mitophagy in WT-MEFs, but not in CerS1$^{top/top}$-MEFs, as detected by the co-localization of lysosomes (LysoTracker green, LTG) and mitochondria (MitoTracker red, MTR) using live cell imaging/fluorescence microscopy or lipidation of LC3B using Western blotting (Fig 1A and B). Reconstitution of WT CerS1 expression, and not catalytically inactive mutant CerS1 (H138A) (Sentelle et al, 2012), restored sodium selenite-mediated mitophagy in CerS1$^{top/top}$-MEFs (Fig 1A and B). Then, to determine whether CerS1/C$_{18}$-ceramide signaling plays any roles in cisplatin-mediated growth inhibition in HPV(+) HNSCC cells, we measured the effects of shRNA-mediated knockdown of CerS1 on the growth of UM-SCC-47 cells in response to cisplatin using MTT assay. Knockdown of CerS1, ~70% downregulation protein compared to scrambled (Scr) shRNA-transfected cells reduced cisplatin-mediated growth inhibition (~eightfold) compared to controls (Fig 1C). These data were also consistent in UM-SCC-47 cells that express siRNA against endogenous CerS1, in which reconstitution of WT CerS1, but not the mutant CerS1 (H208A)-induced mitophagy, detected by decreased ACO2 (aconitase 2), a mitochondrial matrix protein (Fig 1D), which is degraded in response to SS via ceramide-mediated mitophagy, used as a marker for mitophagy (Dany et al, 2016). Restoring WT CerS1 decreased UM-SCC-47 cell proliferation/survival (~40%) compared to vector- or mutant CerS1-transfected cells (Fig 1E). Si-RNA-mediated CerS1 knockdown, and ectopic expression of WT and mutant CerS1 were confirmed by Western blotting (Fig 1F). Thus, these data suggest that CerS1/ceramide signaling plays a key role, at least in part, in cisplatin or SS-induced growth inhibition in HPV(+) UM-SCC-47 cells.

To determine the roles of HPV16-E6 and E7 proteins in HNSCC growth inhibition in response to cisplatin, we measured the effects of siRNA-mediated knockdown of these proteins on the growth of UM-SCC-47 cells in response to cisplatin compared to scrambled

**Figure 1.  HPV-mediated HNSCC cell death is CerS1/ceramide dependent.**

A, B     Involvement of CerS1/ceramide signaling in the induction of mitophagy was assessed using live cell imaging for co-localization of MTR and LTG (A) or Western blotting for induction of LC3 lipidation (B) in MEFs isolated from WT or CerS1$^{top/top}$ mice in the absence/presence of known mitophagy inducer sodium selenite (SS, 10 μM, for 3 h). Effects of ectopic expression of WT CerS1 versus mutant CerS1 (H208A) on mitophagy were also measured by MTR/LTG co-localization (A). Expression of WT CerS1 and H208A CerS1 was confirmed by Western blotting using anti-FLAG antibody (right panel). In (A), images were quantified by ImageJ (shown below each panel), and scale bars represent 100 μm. Data shown represent three independent experiments.

C          Effects of siRNA-mediated knockdown of CerS1 on cisplatin-mediated growth inhibition were measured by MTT assay. Data are means ± SD from three independent experiments, analyzed by unpaired Student's t-test (*P = 0.0006).

D, E     Effects of reconstitution of WT CerS1 (CS) versus mutant CerS1 (H138A) on ACO2 degradation (mitophagy marker) and on growth inhibition in response to SS (10 μM, 3 h) were determined using Western blotting (D) and trypan blue exclusion assay (E) in HPV(+) UM-SCC-47 cells, which were transfected with shRNA that targeted endogenous CerS1. Data represent three independent experiments, analyzed by unpaired Student's t-test and *P = 0.0167 (n = 3).

F          siRNA-mediated CerS1 knockdown (left panel) and ectopic expression of WT and mutant CerS1 (right panel) in (D and E) were confirmed by Western blotting.

Source data are available online for this figure.

(Scr)-siRNA-transfected controls. Knockdown of E6 and E7 proteins confirmed by Western blotting resulted in ~eightfold resistance to cisplatin-mediated growth inhibition in UM-SCC-47 cells compared to controls (Fig 2A and B). To determine whether HPV-E6 versus E7 plays distinct roles in the regulation of growth inhibition in response to chemotherapy, we ectopically expressed these proteins selectively (confirmed by Western blotting) in HPV(−) UM-SCC-22A cells, and measured their effects on growth inhibition or mitophagy in

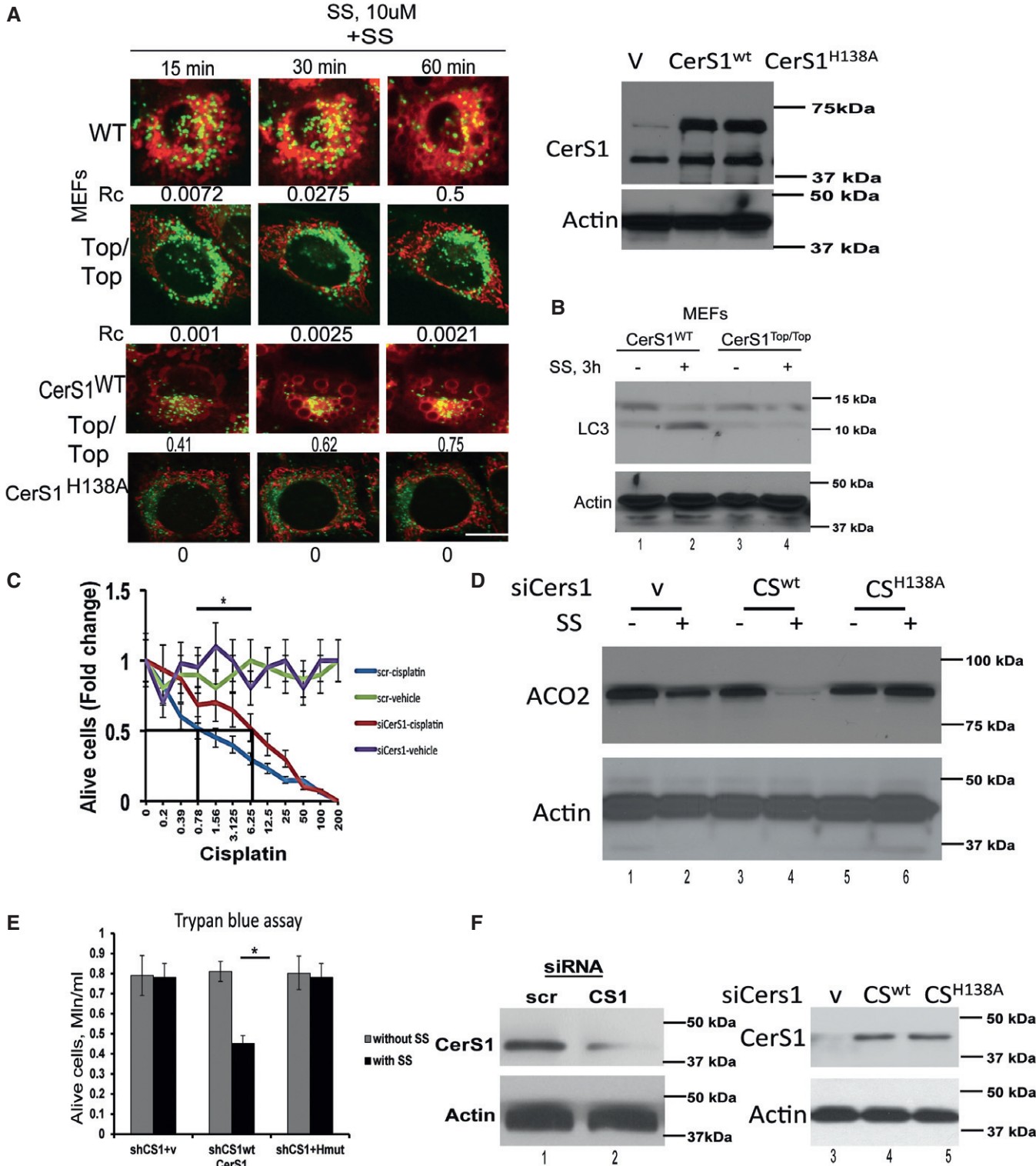

Figure 1.

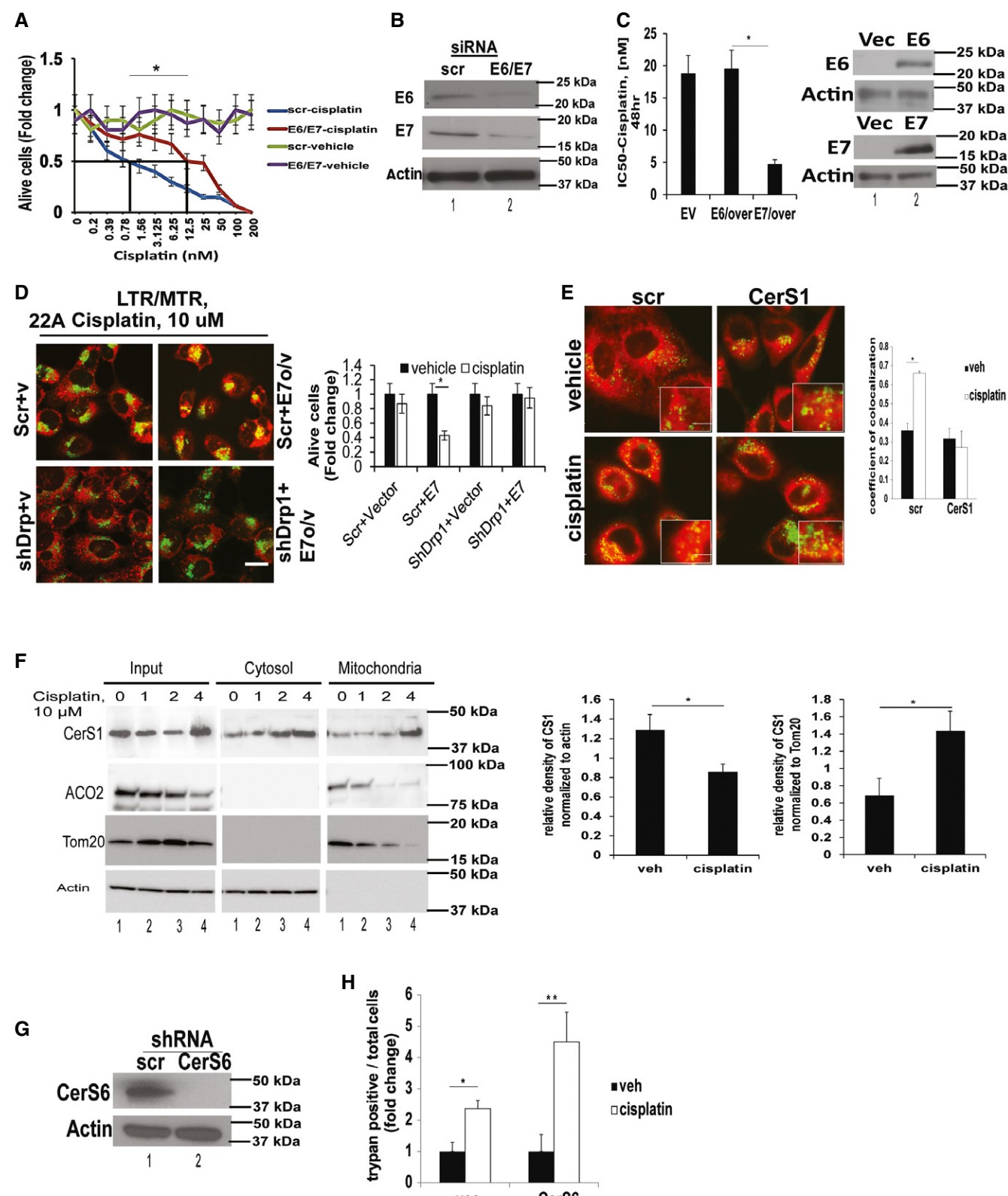

**Figure 2.**

response to cisplatin. Ectopic expression of HPV-E7 sensitized UM-SCC-22A cells to cisplatin (~4.5-fold) compared to vector- or HPV-E6-transfected cells (Fig 2C).

To confirm that HPV-E7-mediated suppression of HNSCC growth/proliferation was dependent on mitophagy, we measured the effects of shRNA-mediated knockdown of Drp1 (dynamin-related

**Figure 2.  HPV-E7 enhances chemotherapy-mediated cellular stress and CerS1/ceramide-mediated lethal mitophagy.**

A, B    Effects of siRNA-mediated knockdown of HPV-E6/E7 on HPV(+) UM-SCC-47 growth inhibition in response to cisplatin at various concentrations for 48 h were assessed by MTT compared to Scr-siRNA-transfected and/or vehicle-treated controls. Successful knockdown of HPV-E6/E7 proteins was confirmed by Western blot analysis (B). Data are means ± SD from three independent experiments, analyzed by unpaired Student's *t*-test (*$P$ = 0.0005).

C       Effects of ectopic expression of HPV-E6 versus HPV-E7 on HPV(−) UM-SCC-22A growth inhibition in response to cisplatin (48 h) were measured by MTT. HPV-E6/E7 protein abundance was measured by Western blotting (right panel). Data are means ± SD from three independent experiments, analyzed by unpaired Student's *t*-test (*$P$ = 0.0042).

D       Effects of shRNA-mediated knockdown of Drp1 on mitophagy induction or growth inhibition were determined using live cell imaging for MTR/LTG co-localization (left panel) or trypan blue exclusion assay (right panel) in UM-SCC-1A, which were transfected with vector-only or HPV-E7. Data shown are means ± SD from three independent experiments, analyzed by unpaired Student's *t*-test (*$n$ = 3, *$P$ = 0.0005). In (D), scale bars represent 100 μm.

E       Effects of shRNA-mediated knockdown of CerS1 on mitophagy in response to cisplatin (48 h) were measured by live cell imaging/confocal micrographs of UM-SCC-47 cells stained with LTG and MTR. Scr-shRNA-transfected and/or vehicle-treated cells were used as controls. Images were quantified by ImageJ, and scale bars represent 100 μm. Data are means ± SD from three independent experiments, analyzed by unpaired Student's *t*-test (*$P$ = 0.00044).

F       Cytoplasmic versus mitochondrial localization of CerS1 in the presence/absence of cisplatin (10 μM) at 0, 1, and 4 h was measured by Western blotting using anti-CerS1 antibody in cytoplasm (cyto)- versus mitochondria (mito)-enriched cellular fractions of UM-SCC-47 cells. Actin and Tom20 or ACO2 were used as controls for cytoplasm- and mitochondria-enriched fractions, respectively. In Western blot panels, images are representative of three independent experiments. Western blot images represent three independent studies, and images were quantified using ImageJ and analyzed by unpaired Student's *t*-test (*$n$ = 3, left panel *$P$ = 0.0177, right panel *$P$ = 0.0113).

G, H    Effects of shRNA-mediated knockdown of CerS6, confirmed by Western blotting (G), on UM-SCC-47 growth inhibition in response to cisplatin were measured by trypan blue exclusion assay (H). Data are means ± SD from three independent experiments, analyzed by unpaired Student's *t*-test (*$P$ = 0.037, **$P$ = 0.00013). In Western blots (G), actin was used as a loading control.

Source data are available online for this figure.

protein 1), which was shown to be necessary for ceramide-dependent mitophagy (Sentelle *et al*, 2012; Dany *et al*, 2016) on this process. Ectopic expression of HPV-E7 (Fig EV2B, left panel) enhanced mitophagy (increased MTR/LTG co-localization) and growth suppression in response to cisplatin compared to vector-transfected UM-SCC-1A cells (Fig 2D, left and right panels; Fig EV2A shows the time course from this experiment). Moreover, shRNA-mediated Drp1 knockdown (Fig EV2B, right panel) prevented the effects of HPV-E7 overexpression on cisplatin-induced mitophagy and growth inhibition in UM-SCC-22A cells (Fig 2D, left and right panels). Thus, these data suggest that HPV-E7 enhances growth suppressive effects of cisplatin via, at least in part, Drp1-mediated mitophagy.

ShRNA-mediated knockdown of CerS1 completely abrogated co-localization of LysoTracker green (LTG) and MitoTracker red (MTR) in response to cisplatin (Fig 2E). These data were further confirmed by increased accumulation of CerS1 in mitochondria in response to cisplatin (4 h) compared to controls, using Western blotting in cytoplasm versus mitochondria-enriched cellular fractions (Fig 2F). As an additional control, we also measured the effects of silencing CerS6, which mainly generates $C_{16}$-ceramide, on cisplatin-induced cell death in UM-SCC-47 cells. The data showed that shRNA-mediated knockdown of CerS6, confirmed by Western blotting, had no inhibitory effect on cell death/growth inhibition in response to cisplatin (Fig 2G and H). Thus, these data suggest that HPV-E7 enhances CerS1/ceramide-dependent lethal mitophagy in response to cisplatin.

**Lethal mitophagy is mediated by mitochondria-targeted $C_{18}$-ceramide via HPV-E7 signaling**

To determine whether ceramide accumulation in mitochondria is sufficient to replicate the cellular response to cisplatin, we treated HPV(+) UM-SCC-47 cells with $C_{18}$-pyr-cer. A pyridinium ring conjugated to the sphingosine backbone targets this ceramide analogue to mitochondria (Senkal *et al*, 2006). $C_{18}$-pyr-cer increased mitophagy

detected by decreased oxygen consumption rate, measured using the Seahorse Bioscience XF24 (Beeson *et al*, 2010) compared to vehicle-treated controls (Figs 3A and EV2C). Additionally, using transmission electron microscopy (TEM), we detected decreased number mitochondria and increased number of autophagosomes (suggesting increased mitophagy) in response to $C_{18}$-pyr-cer in HPV (+) UM-SCC-47 compared to HPV(−) UM-SCC-22A cells (Fig 3B). Induction of mitophagy by $C_{18}$-pyr-cer was also measured by live cell imaging using immunofluorescence to detect time-dependent degradation of mitochondria by autophagosomes (Fig 3C and D). SiRNA-mediated knockdown of LC3B or ATG5 prevented $C_{18}$-pyr-cer-mediated mitophagy and cell death compared to Scr-siRNA-transfected controls (Fig 3C–E). LC3B knockdown (confirmed by Western blotting) also resulted in resistance to cisplatin (~threefold) compared to Scr-siRNA-transfected controls (Fig EV3A and B). As an additional control, we measured the effects of conventional $C_{18}$-, $C_{16}$-, and $C_6$-ceramides, which are not targeted to mitochondria, compared to their pyridinium-conjugated analogues, which are known to accumulate in mitochondria (Senkal *et al*, 2006), on mitophagy induction. $C_{18}$-pyr-cer, $C_{16}$-pyr-cer, and $C_6$-pyr-cer, but not their conventional analogues, mediated mitophagy in UM-SCC-47 cells compared to vehicle-treated controls (Fig 3F). Thus, these data suggest that induction of mitochondrial accumulation of $C_{18}$-pyr-cer, independent of fatty acyl chain length, enhances mitophagy-dependent cell death/growth suppression in HPV(+) compared to HPV(−) HNSCC cells. It should be noted that $C_{18}$-pyr-cer induces lethal mitophagy in HPV(−) HNSCC (UM-SCC-22A) cells when used at higher concentrations or longer treatment time points (5–10 μM, 24–48 h), when IC50 is reached (Fig EV1F), as described previously (Sentelle *et al*, 2012).

**Inhibition of Rb by HPV-E7 targets HNSCC mitochondria for ceramide-dependent mitophagy**

To examine a possible role for HPV-E6/E7 oncoproteins in the enhanced response of HPV-positive HNSCC, we knocked down

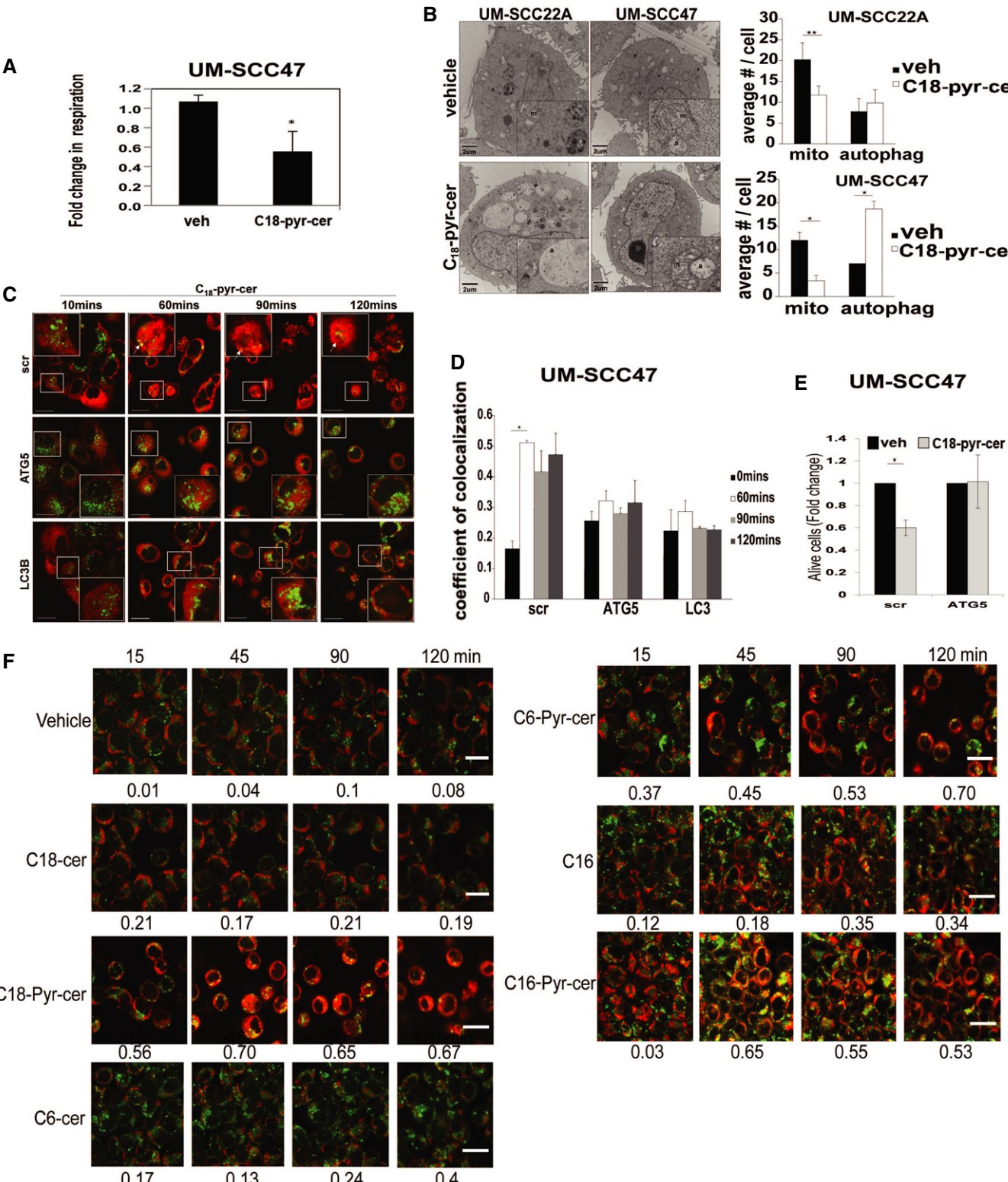

**Figure 3.**

HPV-E6/E7 in HPV-positive cells and examined response to treatment with $C_{18}$-pyr-cer. We found that knockdown of E6/E7 resulted in resistance to $C_{18}$-pyr-cer-induced cell death compared to Scr-siRNA-transfected controls (Fig EV4A). Moreover, siRNA-mediated

knockdown of E6/E7 attenuated $C_{18}$-pyr-cer-induced mitophagy, measured by change in oxygen consumption rate (OCR), and co-localization of LTG and MTR (Fig 4A and B). However, ectopic expression of E7, but not E6, resulted in increased cell death in

**Figure 3.  Mitochondrial targeting of ceramide induces lethal mitophagy in HPV(+) cells.**

A    Mitophagy was measured in response to $C_{18}$-pyr-cer (20 μM, for 2 h) in UM-SCC-47 cells using Seahorse analyzer for the detection of mitochondrial respiration (oxygen consumption rate) compared to vehicle-treated controls. Data are means ± SD from three independent experiments, analyzed by unpaired Student's t-test (*P = 0.00002).

B    Effects of $C_{18}$-pyr-cer on the induction of mitophagy in HPV(−) UM-SCC-22A versus HPV(+) UM-SCC-47 cells were measured by TEM compared to vehicle-treated controls (left panel). Number of mitochondria (mito) and autophagosomes (autophag) were counted in TEM micrographs (right panel). Data are means ± SD from three independent experiments, analyzed by unpaired Student's t-test (**P = 0.00082, *P = 0.0121).

C, D    Mitophagy was detected in response to $C_{18}$-pyr-cer (20 μM) at 0- to 120-min exposure using live cell imaging confocal micrographs of cells stained with LTG and MTR in the presence of shRNA-mediated knockdown of ATG5 or LC3B compared to Scr-shRNA-transfected controls. Images were quantified using ImageJ (D). In (C), upper panel, white arrows show the degradation of mitochondria in response to $C_{18}$-pyr-cer-mediated mitophagy in a time-dependent manner. Data are means ± SD from three independent experiments, analyzed by unpaired Student's t-test (n = 3, *P = 0.00031), and scale bars represent 100 μm.

E    UM-SCC-47 growth inhibition was measured by trypan blue exclusion assay in the presence of Scr-siRNA or ATG5-siRNA in response to $C_{18}$-pyr-cer exposure (20 μM, 48 h). Data are means ± SD from three independent experiments, analyzed by unpaired Student's t-test (n = 3, *P = 0.0041).

F    Mitophagy induction in UM-SCC-47 cells in response to mitochondria-targeted $C_{18}$-pyr-cer, $C_{16}$-pyr-cer, or $C_6$-pyr-cer versus conventional $C_{18}$-ceramide, $C_{16}$-ceramide, or $C_6$-ceramide ($C_{18}$-, $C_{16}$, $C_6$-cer) was measured by live cell imaging using MTR/LTG colocalization (15–120 min, 10 μM). Images were quantified using ImageJ (shown below each panel). Images represent three independent studies, and scale bars represent 100 μm.

response to $C_{18}$-pyr-cer in HPV(−) UM-SCC-22A cells, and enhanced mitophagy, measured by oxygen consumption rate (OCR) using the Seahorse analyzer, or co-localization of LTG and MTR, in response to $C_{18}$-pyr-cer (Figs EV4B and 4C–E). Taken together, these data suggest that HPV-E7 plays a key role in sensitizing HNSCC cells to treatment with either cisplatin or $C_{18}$-pyr-cer.

To further confirm that HPV-E7, but not HPV-E6, plays an important role in sensitizing HNSCC to $C_{18}$-pyr-cer, we knocked down the E6 target p53 or the E7 target RB, in HPV-negative cells and measured their effects on cell death. The data demonstrated that shRNA-mediated knockdown of RB, but not p53, increased $C_{18}$-pyr-cer-induced cell death (Fig 5A) compared to Scr-shRNA-transfected and vehicle-treated controls. Next, we utilized a vector to ectopically express a mutant RB (RB10), which is catalytically active but has a mutation in the E7 binding domain (L-X-C-X-E) so that it remains active in E7-expressing HPV(+) cells (Dick *et al*, 2000). Ectopic expression of RB10 in UM-SCC-47 cells attenuated cell death and prevented mitophagy (measured by decreased oxygen consumption rate, and increased co-localization of LTG and MTR) in response to $C_{18}$-pyr-cer compared to controls (Fig 5B–D). Thus, these data suggest that inhibition of RB signaling by HPV-E7 is key for regulation of ceramide-dependent lethal mitophagy.

### Activation of E2F5 via inhibition of RB by HPV-E7 enhances ceramide-induced mitophagy

To identify the downstream mediators of HPV-E7/RB signaling in enhancing ceramide-mediated lethal mitophagy, we investigated the roles of E2F proteins, the canonical downstream targets of RB. Because E2F1, 4, and 5 proteins have been associated with autophagy, cell death, or inhibition of cell growth previously (Polager *et al*, 2008; Jiang *et al*, 2010; Morales *et al*, 2014), we assessed the effects of shRNA-mediated knockdown of E2F1, E2F4, or E2F5 on ceramide-induced mitophagy. The data showed that knockdown of E2F5, but not E2F1 or E2F4, attenuated cell death in response to $C_{18}$-pyr-cer in HPV(+) HNSCC cells (Fig 6A, left panel). Efficiency of shRNA-mediated knockdown of E2F1, E2F4, and E2F5 mRNAs compared to Scr-shRNA-transfected cells was confirmed using RT–PCR (Fig 6A, right panel). The involvement of E2F5 in HPV-E7-mediated mitophagy was also consistent with a strong association between RB and E2F5, measured by proximity ligation assay (PLA) using fluorescently labeled anti-RB and anti-E2F5 antibodies, which

was enhanced by knockdown of HPV-E6/E7 (Fig 6B). Moreover, shRNA-mediated knockdown of E2F5 attenuated mitophagy (colocalization of LTG and MTR) in response to $C_{18}$-pyr-cer in HPV(+) UM-SCC-47 cells (Fig 6C). Conversely, expression of exogenous E2F5 (confirmed by Western blotting, Fig 6D, lower panel) in HPV(−) UM-SCC-22A cells enhanced cell death and increased mitophagy (colocalization of LTG and MTR) in response to $C_{18}$-pyr-cer compared to vector-only-transfected and vehicle-treated controls (Fig 6D and E). Thus, these data suggest a role for E2F5 activation via the inhibition of RB by HPV-E7 in enhancing ceramide-mediated lethal mitophagy.

### HPV-E7/ceramide-mediated mitophagy is induced by E2F5-Drp1 complex

To further define the mechanism by which E2F5 enhances HPV-E7/ceramide-mediated mitophagy, we investigated the involvement of dynamin-related protein 1 (Drp1) in this process, due to increased mitochondrial fission observed in TEM micrographs containing images of $C_{18}$-pyr-cer-treated UM-SCC-47 cells (Fig 3B, bottom right image). Drp1 oligomerization and/or activation were observed in HPV(+) cells treated with cisplatin or $C_{18}$-pyr-cer (Fig 7A). Expression of an inactive/dominant-negative mutant of Drp1 (K38A) inhibited cell death in response to cisplatin or $C_{18}$-pyr-cer compared to vector-transfected and vehicle-treated controls (Fig 7B and C). ShRNA-mediated knockdown of E2F5 prevented $C_{18}$-pyr-cer-induced Drp1 oligomerization/activation compared to Scr-shRNA-transfected and vehicle-treated controls (Fig 7C). Thus, these data suggest that Drp1 plays a key role in HPV-E7/ceramide-mediated lethal mitophagy.

Interestingly, our data also demonstrated that alterations of E2F5 abundance by shRNA transfections had no effect on Drp1, LC3, or ATG5 mRNAs compared to controls (Fig EV5A), suggesting that canonical transcription factor function of E2F5 in HPV-E7/ceramide-mediated mitophagy might be dispensable. This was supported by the cytoplasmic localization of E2F5 in the absence/presence of $C_{18}$-pyr-cer in HPV(+) UM-SCC-47 cells (Fig EV5B and C). These data suggested to us a novel hypothesis that cytoplasmic E2F5 might associate with Drp1 for activation, leading to increased mitochondrial fission in response to ceramide stress in HPV(+) HNSCC cells. Indeed, increased Drp1-E2F5 association was detected in response to $C_{18}$-pyr-cer compared to controls, measured by PLA using fluorescently labeled anti-Drp1 and anti-E2F5 antibodies (Fig 7D).

    

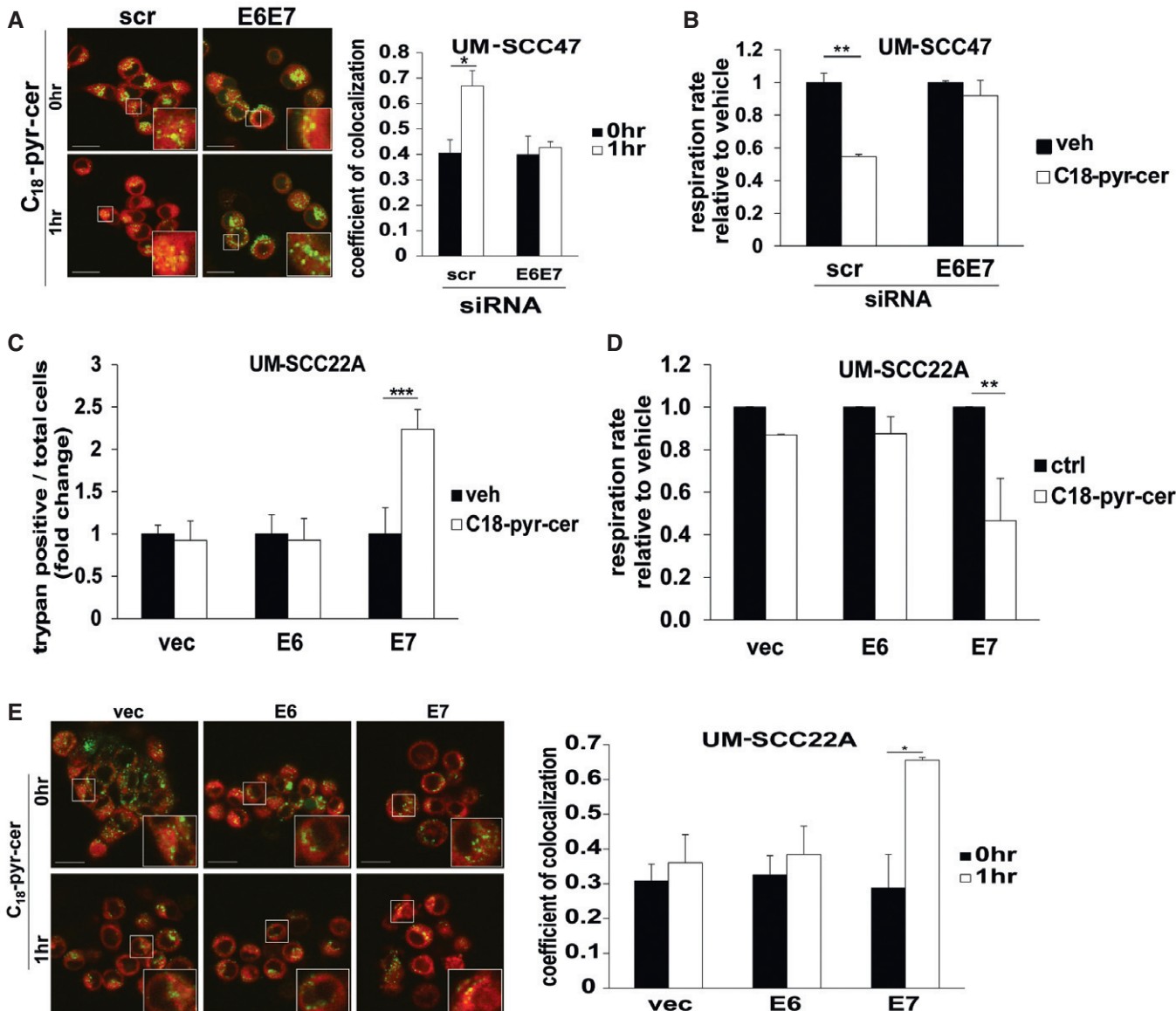

**Figure 4. Mitochondrial ceramide is involved in HPV-E7-mediated lethal mitophagy.**

A, B  Effects of siRNA-mediated knockdown of HPV-E6/E7 on mitophagy in response to $C_{18}$-pyr-cer (20 μM, 1 h) were measured by live cell imaging confocal micrographs of UM-SCC-47 cells stained with LTG and MTR (A) or by mitochondrial respiration (oxygen consumption rate) using Seahorse analyzer (B). Scr-siRNA-transfected and/or vehicle-treated cells were used as controls. Data are means ± SD from three independent experiments, analyzed by unpaired Student's *t*-test (*n* = 3, *P = 0.021, **P = 0.0004). In (A), images were quantified using ImageJ, and scale bars represent 100 μm.

C–E  Roles of ectopic expression of HPV-E6 or E7 in UM-SCC-22A cells for enhancing mitochondrial $C_{18}$-pyr-cer-mediated cell death and mitophagy were measured by trypan blue exclusion assay (C) and Seahorse analyzer for measuring mitochondrial respiration (oxygen consumption rate) (D) or using live cell imaging confocal micrographs of UM-SCC-22A cells stained with LTG and MTR (E). Vector-transfected and/or vehicle-treated cells were used as controls. Data are means ± SD from three independent experiments, analyzed by unpaired Student's *t*-test (*n* = 3, *P = 0.0093, **P = 0.0072, ***P = 0.00026). In (E), images were quantified using ImageJ, and scale bars represent 100 μm.

Co-immunoprecipitation experiments in HPV-positive UM-SCC-47 cells treated with vehicle control or $C_{18}$-pyr-cer confirmed increased association between Drp1 and E2F5 (Fig 7E). Knockdown of E6/E7 attenuated Drp1-E2F5 association (detected by PLA) compared to Scr-siRNA-transfected HPV(+) UPI-SCC-90 cells (Fig 7F). Knockdown of HPV-E7 was confirmed by increased pRB and decreased E7 protein abundance, and knockdown of HPV-E6 was confirmed by increased p53 protein abundance compared to controls in UPI-SCC-90 cells (Fig 7F, right panel). Thus, these data suggest a non-canonical cytoplasmic function for E2F5 to associate with Drp1 upon ceramide stress in response to $C_{18}$-pyr-cer or cisplatin, to induce HPV-E7-mediated lethal mitophagy.

To determine the molecular details of Drp1-E2F5 complex, we identified a specific domain of E2F5 involved in Drp1 interaction by

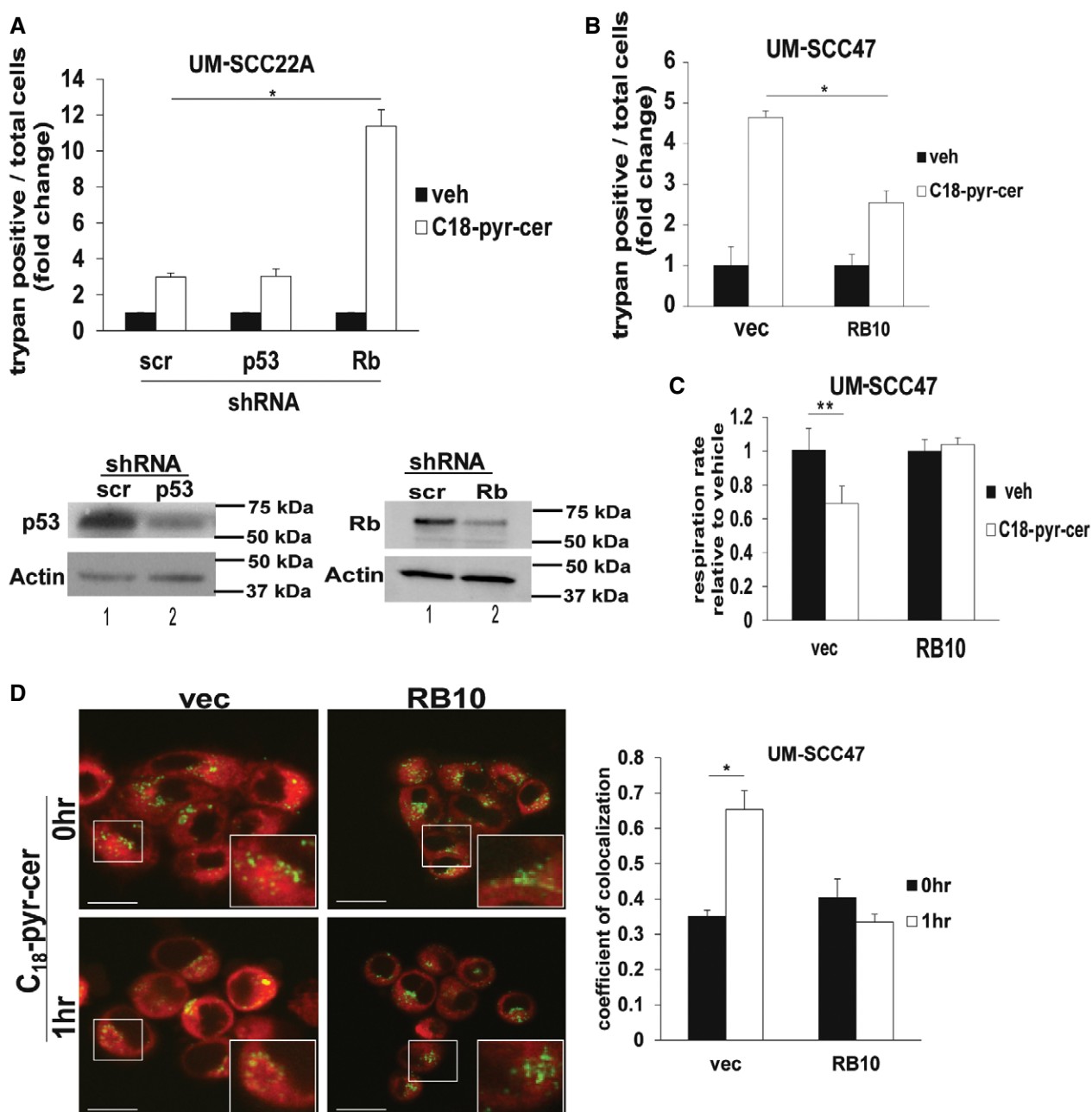

**Figure 5. HPV-E7 targets RB for induction of ceramide-dependent mitophagy.**

A    Roles of shRNA-mediated knockdown of p53, a target of HPV-E6, versus RB, a target of HPV-E7, on ceramide-mediated UM-SCC-22A growth inhibition in response to $C_{18}$-pyr-cer (20 μM, 48 h) were measured by trypan blue exclusion assay. Scr-shRNA-transfected and/or vehicle-treated cells were used as controls. Data are means ± SD from three independent experiments, analyzed by two-way ANOVA ($n = 3$, *$P = 1.7 \times 10^{-6}$). Protein abundance of p53 and RB was measured by Western blotting in cells transfected with shRNAs. Actin was used as a loading control (lower panels).

B–D    Effects of ectopic expression of RB10 (active mutant of RB, which is not recognized by HPV-E7) on UM-SCC-47 growth inhibition and mitophagy were measured by trypan blue exclusion assay (B) and Seahorse analyzer for detecting mitochondrial respiration (oxygen consumption rate) (C) or using live cell imaging confocal micrographs of UM-SCC-47 cells stained with LTG and MTR (D) in the absence or presence of $C_{18}$-pyr-cer (20 μM, 1 h). Vector-transected cells were used as controls. Data are means ± SD from three independent experiments, analyzed by two-way ANOVA (B, $n = 3$, *$P = 0.0033$) or unpaired Student's $t$-test (C and D, $n = 3$, *$P = 0.0085$, **$P = 0.0084$). In (D), images were quantified using ImageJ, and scale bars represent 100 μm.

Source data are available online for this figure.

molecular docking using the ZDOCK server. These data suggested that the association between Drp1 and E2F5 might involve the "dimerization domain" of E2F5, where E2F5 is known to bind the

activating dimerization partner (DP) protein, between residues 84 and 177 (Fig 8A) (Apostolova *et al*, 2002). To validate these studies, we generated a mutant E2F5, in which the dimerization domain was

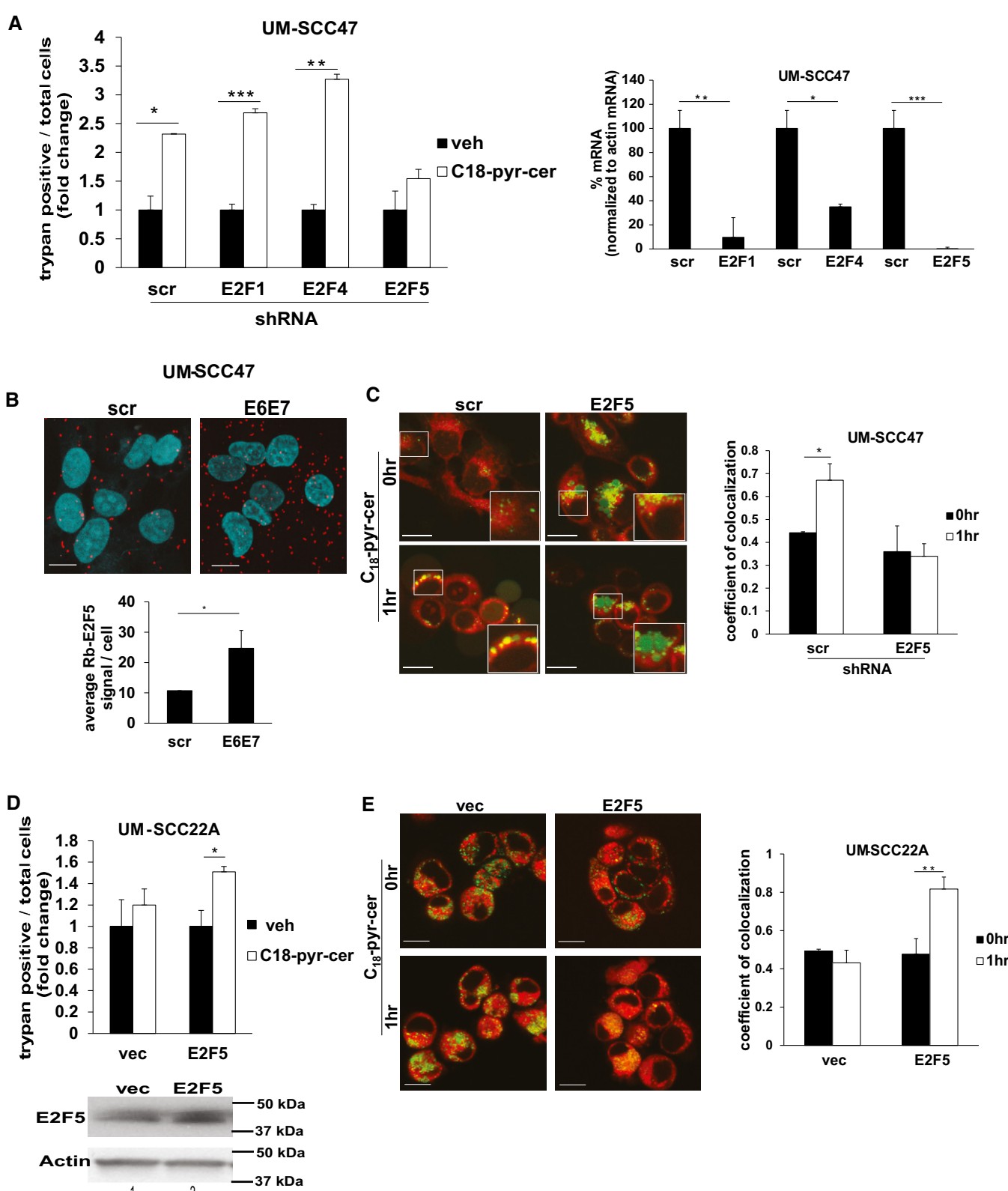

**Figure 6.**

deleted E2F5$^{\Delta84-177}$-GFP. To verify the integrity and correct folding of E2F5$^{\Delta84-177}$-GFP, we measured its association with RB by PLA using labeled anti-GFP and anti-RB antibodies compared to E2F5$^{WT}$-GFP-RB

association. The data demonstrated that E2F5$^{\Delta84-177}$-GFP binds RB as effectively as E2F5$^{WT}$ (Fig 8B). Importantly, deletion of the dimerization domain of E2F5 almost completely inhibited the

**Figure 6.  HPV-E7/RB axis regulates ceramide-dependent mitophagy via E2F5 activation.**

A    UM-SCC-47 [HPV(+)] cell death was measured using trypan blue exclusion assay (trypan blue-positive cells) following shRNA-mediated knockdown of E2F1, E2F4, or E2F5 in response to $C_{18}$-pyr-cer (20 μM, 48 h) compared to Scr-shRNA-transfected and/or vehicle-treated controls. Efficiency of shRNA-mediated knockdown of E2F1, E2F4, or E2F5 mRNAs was measured by RT–PCR (normalized to actin mRNA) compared to Scr-shRNA-transfected controls (right panel). Data are means ± SD from three independent experiments, analyzed by unpaired Student's t-test ($n$ = 3, left panel *$P$ = 0.00025, **$P$ = 0.00018, ***$P$ = 5.4 × $10^{-5}$; right panel *$P$ = 0.0156, **$P$ = 0.0147. ***$P$ = 0.00118).

B    Effects of siRNA-mediated knockdown of HPVE6/E7 on endogenous RB-E2F5 association in UM-SCC-47 cells were measured by PLA using labeled anti-RB and anti-E2F5 antibodies using immunofluorescence (scale bars represent 100 μm). Quantification of PLA signals in images was performed as described by the manufacturer using the PLA software. Data are means ± SD from three independent experiments, analyzed by unpaired Student's t-test ($n$ = 3, *$P$ = 0.040).

C    Effects of shRNA-mediated E2F5 knockdown on mitophagy in response to vehicle versus $C_{18}$-pyr-cer (20 μM, 1 h) were measured by live cell imaging confocal micrographs of UM-SCC-47 cells stained with LTG and MTR (scale bars represent 100 μm). Images were quantified using ImageJ, and data are means ± SD from three independent experiments, analyzed by unpaired Student's t-test (*$P$ = 0.023).

D–E    Cell death (trypan blue-positive cells) or mitophagy was measured in response to ectopic expression of E2F5 in the presence/absence of $C_{18}$-pyr-cer (20 μM, 48 h) by trypan blue exclusion assay (D) or by live cell imaging (1 h) confocal micrographs of UM-SCC-22A cells stained with LTG and MTR (E). Vector-transfected cells were used as controls. Images were quantified using ImageJ (scale bars represent 100 μm), and data are means ± SD from three independent experiments, analyzed by unpaired Student's t-test ($n$ = 3, *$P$ = 0.0084, **$P$ = 0.0012). Protein abundance of E2F5 after transient transfections was measured by Western blotting. Actin was used as a loading control (D, lower panel).

Source data are available online for this figure.

association between Drp1 and E2F5$^{\Delta 84–177}$-GFP, whereas ectopically expressed E2F5$^{WT}$-GFP showed increased Drp1-E2F5$^{WT}$ interaction in HPV(−) cells, measured by PLA (Fig 8C, left panel). Comparable transfection efficiencies of GFP-tagged E2F5 proteins compared to GFP-alone (vector control) were measured by immunofluorescence in these cells (Fig 8C, right panel). Ectopic expression of E2F5$^{WT}$, but not E2F5$^{\Delta 84–177}$, enhanced ceramide-induced cell death in HPV (−) cells (Fig 8D) and increased mitophagy (as measured by colocalization of LTG and MTR) (Fig 8E). Thus, these data suggest that the dimerization domain of E2F5 within amino acids 84–177 is involved in Drp1 association, leading to increased mitochondrial fission and ceramide-dependent mitophagy by HPV-E7 signaling.

It is known that Drp1 engages with its mitochondrial receptor MFF to induce mitochondrial fission and mitophagy (Koirala *et al*, 2013). Thus, to determine the mechanism by which E2F5-Drp1 association enhances HPV-E7/ceramide-induced lethal mitophagy, we examined the mitochondrial localization of Drp1 and its association with mitochondrial receptor MFF. The data showed that $C_{18}$-pyr-cer induced Drp1-MFF association, which was attenuated by stable knockdown of E2F5 in UM-SCC47 cells compared to controls (Fig 9A). Conversely, ectopic expression of E2F5$^{WT}$ in HPV-negative UM-SCC-1A cells enhanced Drp1-MFF association in response to $C_{18}$-pyr-cer compared to vector-only-transfected and vehicle-treated controls (Fig 9B). These data were confirmed by immunoprecipitation/Western blotting, which showed that shRNA-mediated knockdown of E2F5 in UM-SCC-47 cells prevented Drp1-MFF interaction compared to controls in response to $C_{18}$-pyr-cer (Fig 9C, right panel, lanes 4 and 2). In reciprocal studies, ectopic expression of E2F5 in UM-SCC-22A cells enhanced the association between Drp1 and MFF compared to vector-transfected controls in response to C18-pyr-cer (Fig 9D, right panel, lanes 4 and 2). In addition, $C_{18}$-pyr-cer had no effect on the association between Drp1 and SMCR7/MiD49, mitochondrial dynamics protein (49 kDa), used as an additional control for immunoprecipitation/Western blotting (Fig 9D, right panel). Moreover, $C_{18}$-pyr-cer induced recruitment of Drp1 to mitochondria, measured by Western blotting using mitochondria-enriched subcellular fractions, which was attenuated in response to stable E2F5 knockdown in HPV(+) cells compared to Scr-shRNA-expressing cells (Fig 9E). Reconstitution of E2F5$^{WT}$, but not the E2F5$^{\Delta 84–177}$ mutant in HPV(+) cells, which express stable shRNAs against endogenous

E2F5, rescued Drp1 recruitment to mitochondria in response to ceramide stress compared to vector-transfected controls (Fig 9F). Thus, these data suggest that E2F5 enhances HPV-E7/ceramide-mediated mitophagy by inducing activation and recruitment of Drp1 to mitochondria for MFF-mediated mitochondrial fission.

## Reconstitution of E2F5-Drp1 association by E2F5-peptide mimetic enhances mitophagy in HPV(−) cells

To determine whether the putative Drp1-binding domain of E2F5 alone was sufficient to reconstitute E2F5 activity for mitophagy induction in HPV-negative cells, we generated a peptide based on the minimal stretch of amino acids, corresponding to Drp1 binding E2F5 residues (146–175, biotin-RRRRRRRR-ELDQQKLWL QQSIKNVMDDSINNRFSYVTHED) and scrambled control peptide, which contains the same amino acids as E2F5-pept in randomized/ scrambled order (biotin-RRRRRRRR-LILFVIKLHQDVNDMRNSNQD QTQSE-DRESKWY), as predicted by molecular docking studies. Eight arginine residues (R8) were included on the N-terminus to enhance cell penetration of the E2F5 peptide (Raucher & Ryu, 2015). Importantly, treatment of UM-SCC-22A cells with the E2F5-pept largely increased cisplatin or $C_{18}$-pyr-cer-induced cell death compared to scr-pept (Fig 10A and B), which was attenuated by shRNA-mediated Drp1 knockdown (Fig 10C). E2F5-pept also enhanced $C_{18}$-pyr-cer-induced mitophagy, measured by increased colocalization of LTG and MTR (Fig 10D). Moreover, E2F5-pept but not scr-pept enhanced ceramide-induced Drp1 recruitment to mitochondria in HPV(−) cells (Fig 10E). In addition, pharmacologic inhibition of Drp1 using Mdivi (Cassidy-Stone *et al*, 2008) prevented $C_{18}$-pyr-cer-mediated growth suppression or ACO2 degradation with/without E2F-peptide mimetic (+/− p) in UM-SCC-47 or UM-SCC-22A cells (Fig 10F and G). Thus, these data suggest that activation of Drp1 is key for $C_{18}$-pyr-cer and E2F-peptide-mediated HNSCC growth suppression.

To assess whether increased E2F5-mediated Drp1 activation increases cisplatin-mediated tumor suppression, we generated HPV (−) UM-SCC-22A-derived xenograft tumors in SCID mice, and measured the effects of E2F5-pept or scr-pept on tumor growth in the absence/presence of cisplatin (3.5 mg/kg every 3 days for 2 weeks). Interestingly, treatment with E2F5-pept almost completely

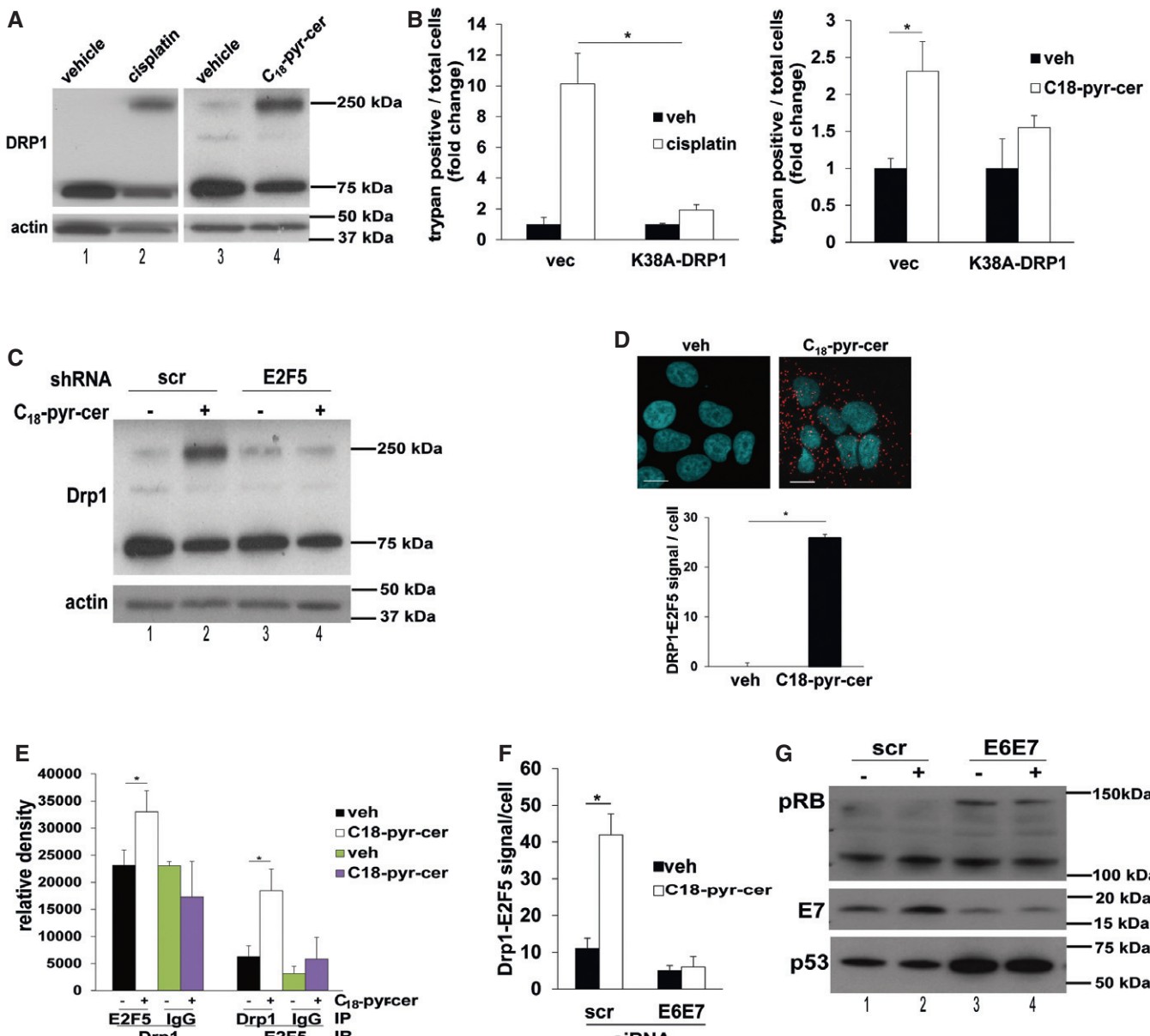

**Figure 7. E2F5/Drp1 complex enhances HPV-E7/ceramide-dependent lethal mitophagy.**

A    UM-SCC47 cells were treated with cisplatin or $C_{18}$-pyr-cer, and Western blotting was performed with cell lysates in the absence of reducing agents in the lysis buffer to detect monomeric and dimeric Drp1 protein abundance compared to vehicle-treated controls. Actin was used as a loading control. Images represent three independent studies.

B    Cell death was assessed by trypan blue exclusion assay in UM-SCC47 cells expressing dominant-negative Drp1 mutant (K38A-DRP1) and treated with 40 μM cisplatin (left panel) or 20 μM $C_{18}$-pyr-cer (right panel) for 48 h compared to vector-transfected controls. Data are means ± SD from three independent experiments, analyzed by two-way ANOVA (left panel, $n = 3$, *$P = 0.0048$; right panel, $n = 3$, *$P = 0.0015$).

C    Effects of E2F5 knockdown on Drp1 oligomerization were measured by Western blotting compared to Scr-shRNA-transfected cells in the absence/presence (−/+) of $C_{18}$-pyr-cer. Actin was used as a loading control. Data represent three independent experiments.

D, E    Association between E2F5 and Drp1 in response to ceramide stress ($C_{18}$-pyr-cer at 10 μM for 1 h) was measured in UM-SCC-47 cells using PLA (D) or immunoprecipitation followed by Western blotting (E) with anti-Drp1 or anti-E2F5 antibodies (IgG was used as a control). Data are means ± SD from three independent experiments, analyzed by unpaired Student's *t*-test ($n = 3$, *$P = 0.037$). In (D), PLA images were quantified using the PLA software as described by the manufacturer and analyzed by unpaired Student's *t*-test ($n = 3$, *$P = 1.3 \times 10^{-5}$) and scale bars represent 100 μm.

F    Effect of siRNA-mediated knockdown of HPVE6/E7 on Drp1-E2F5 association in the absence/presence of ceramide stress ($C_{18}$-pyr-cer, 10 μM, 1 h) was measured by PLA using fluorescently labeled anti-Drp1 and anti-E2F5 antibodies in HPV(+) UPI-SCC-90 cells. PLA signals were quantified as described by the manufacturer using the PLA quantification software. Data are means ± SD from three independent experiments, analyzed by unpaired Student's *t*-test ($n = 3$, *$P = 0.010$).

G    Effects of siRNA-mediated HPV-E6/E7 knockdown on pRB, HPV-E7, and p53 protein abundance were confirmed by Western blotting compared to Scr-siRNA-transfected controls in the absence/presence (−/+) of $C_{18}$-pyr-cer. Total RB was used as a loading control (Rb band, upper panel).

Source data are available online for this figure.

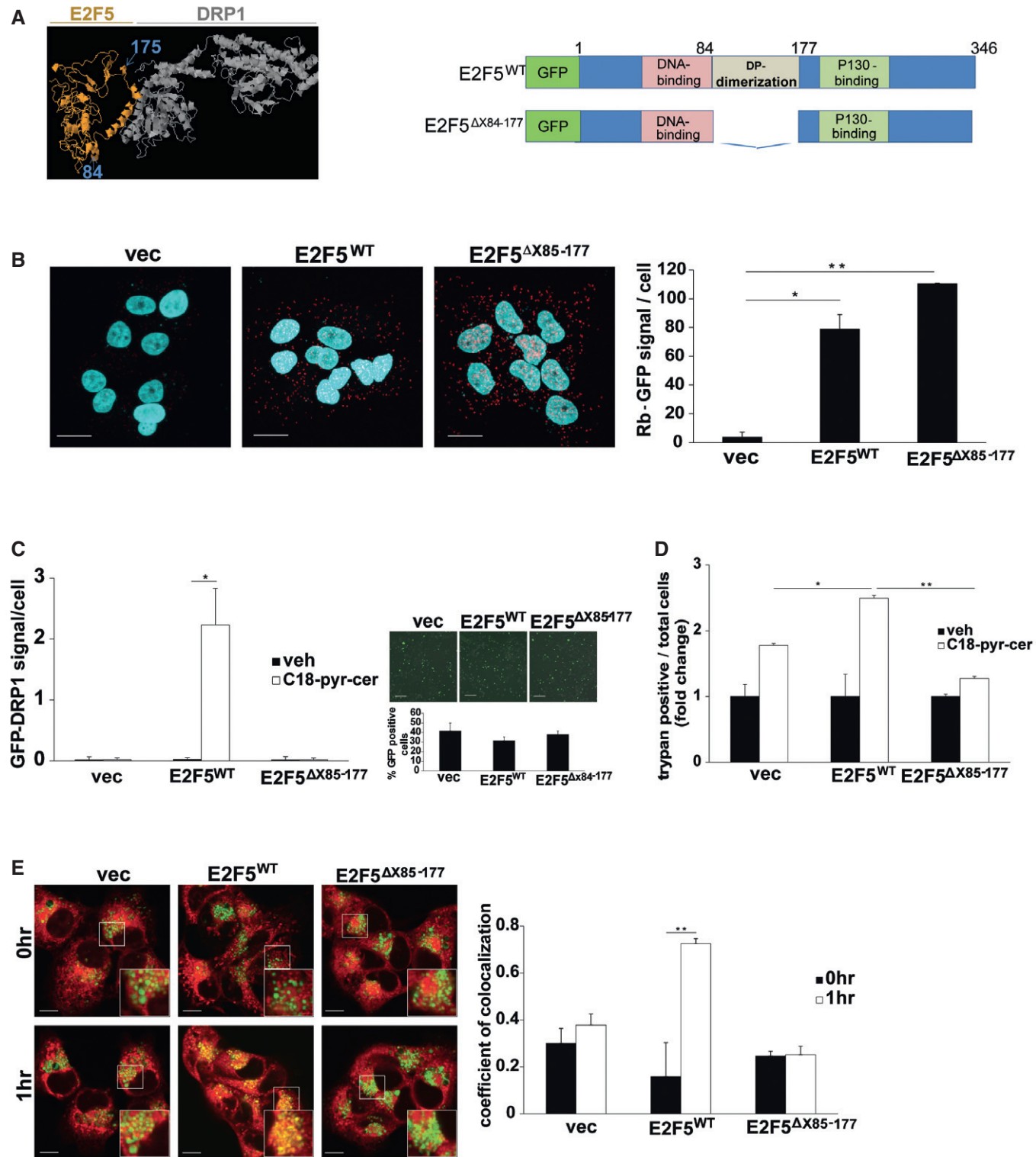

Figure 8.

inhibited tumor growth while control peptide treatment had no effect (Fig 10H). These data were consistent with increased mitophagy (measured by TEM), co-localization of Drp1 and E2F5 (measured by PLA), and ACO2 degradation (measured by Western blotting) in tumors treated with E2F5-pept in response to cisplatin compared to control tumors treated with Scr-pept with/without cisplatin (Fig 10I–K). Thus, these data suggest that induction of E2F5-Drp1 complex using E2F5-pept is associated with decreased tumor growth in response to cisplatin compared to controls (tumors isolated from Scr-pept/vehicle-, Scr-pept/cisplatin-, and/or E2F5-

**Figure 8. Activation of Drp1 by E2F5-Drp1 complex enhances HPV-E7/ceramide-mediated lethal mitophagy.**

A  Top computer-generated model of E2F5 (orange)-Drp1 (gray) docking is shown, with the first and last residues of the known dimerization domain of E2F5 indicated, including residues 85 and 177 (left panel). Models of GFP-tagged wild-type E2F5 (E2F5$^{WT}$) and dimerization domain deleted mutant of E2F5 (E2F5$^{\Delta 85-177}$) are shown (right panel).

B  UM-SCC22A cells transfected with E2F5$^{WT}$-GFP or E2F5$^{\Delta 84-177}$-GFP were used for PLA to measure their association of RB using labeled anti-RB and anti-GFP antibodies in response to C$_{18}$-pyr-cer (10 μM, 1 h). PLA images (scale bars represent 100 μm) were quantified using the PLA software as described by the manufacturer. Data are means ± SD from three independent experiments, analyzed by unpaired Student's *t*-test (*n* = 3, *P = 0.0049, **P = 0.00031).

C–E  UM-SCC22A cells expressing vector, E2F5$^{WT}$ or mutant E2F5$^{\Delta 85-177}$, were used for the measurement of Drp1-E2F5 association in the absence/presence of C$_{18}$-pyr-cer (10 μM, 1 h) using PLA (C) compared to vector-transfected controls. Transfection efficiency and abundance of ectopically expressed E2F5$^{WT}$-GFP and mutant E2F5$^{\Delta 85-177}$-GFP were detected by immunofluorescence (scale bars represent 100 μm). Vector-GFP was used as a control (right panel, C). Moreover, effects of E2F5$^{WT}$ versus mutant E2F5$^{\Delta 85-177}$ on cell death or mitophagy in response to C$_{18}$-pyr-cer (20 μM, 48 h) were measured by trypan blue exclusion assay (D) or by live cell imaging confocal micrographs (E) of UM-SCC-22A cells stained with LTR and MTDR (MitoTracker deep red, imaged with blue channel, and colored green for imaging). Images were quantified using ImageJ (scale bars represent 100 μm). Data are means ± SD from three independent experiments, analyzed by unpaired Student's *t*-test (C and E, *n* = 3, *P = 0.011, **P = 0.0025) or two-way ANOVA (D, *n* = 3, *P = 0.0142, **P = 1.6 × 10$^{-8}$).

pept/vehicle-treated animals) in HPV(−) HNSCC-xenograft-derived tumors in SCID mice. Thus, these results suggest that the Drp1-binding domain of E2F5 is sufficient to induce mitochondrial localization of Drp1, resulting in enhanced mitochondrial fission and ceramide-dependent lethal mitophagy without the expression of HPV-E7 in HNSCC *in situ* and *in vivo*.

## Discussion

Epidemiologic studies indicate that exposure to HPV increases the risk of developing HNSCC (D'Souza *et al*, 2007; Leemans *et al*, 2011; Lehtinen & Dillner, 2013; Killock, 2015). Interestingly, there is evidence of improved survival outcomes in HPV(+) compared to HPV(−) HNSCC patients (Fakhry *et al*, 2008; Ang *et al*, 2010; Mirghani *et al*, 2015). However, molecular mechanisms of how HPV infection improves sensitivity to chemotherapy in patients with HNSCC have been largely unknown. Here, we sought to uncover mechanisms and specific signaling involved in this process. Our novel data demonstrate that HPV16-E7 selectively mediates HNSCC cell death via activation of cytoplasmic E2F5 function, liberated by

targeting RB. Activated E2F5 then binds and stabilizes Drp1 oligomers as a scaffolding molecule, inducing mitochondrial translocation of Drp1 and mitochondrial fission, leading to ceramide-dependent recruitment of LC3-containing autophagosomes and subsequent mitophagy (Fig 11). Inhibition of E2F5-Drp1 complex formation by site-directed mutagenesis of E2F5 attenuates HPV-E7/ceramide-mediated lethal mitophagy, whereas restoring Drp1 activation by ectopic expression of HPV-E7 or treatment with E2F5-peptide mimetic (to reconstitute E2F5-pept-Drp1 interaction) enhances ceramide-dependent mitophagy and tumor suppression in HPV(−) HNSCC in culture and in mice.

HPV has shifted the epidemiological landscape and prognosis of HNSCC with strong evidence of improved therapeutic response and survival outcomes when compared to HPV(−) patients (Moody *et al*, 2007; Cosway & Lovat, 2016). HPVs are small circular DNA viruses, which can infect epithelial cells. There are ~150 different HPV types, including cancerous high risk type HPV16, the most common HPV found in HNSCC. HPV16 proteins include late proteins, such as L1 and L2, and early proteins, such as E6 and E7 oncoproteins (Dyson *et al*, 1989; Halbert *et al*, 1992; White *et al*, 2012). E6 binds cellular ubiquitin ligase E6AP and targets p53 for

**Figure 9. E2F5 enhances Drp1 translocation to mitochondria and association with MFF to induce HPV-E7/ceramide-dependent mitophagy.**

A  Effects of stable knockdown of E2F5 using shRNA on DRP1-MFF association with/without C$_{18}$-pyr-cer (10 μM, 1 h) were measured by PLA (scale bars represent 100 μm) using anti-DRP1 and anti-MFF antibodies compared to Scr-shRNA-transfected UM-SCC-47 controls. PLA fluorescence images were quantified using the PLA software as described by the manufacturer. Data are means ± SD from three independent experiments, analyzed by unpaired Student's *t*-test (*n* = 3, *P = 0.0042).

B  HPV(−) UM-SCC-22A cells transiently transfected with exogenous E2F5 or empty vector (vec) were used to measure the association between Drp1 and MFF in the absence/presence of C$_{18}$-pyr-cer (10 μM, 1 h) by PLA (scale bars represent 100 μm) using anti-DRP1 and anti-MFF antibodies. PLA fluorescence images were quantified using the PLA software as described by the manufacturer. Data are means ± SD from three independent experiments, analyzed by unpaired Student's *t*-test (*n* = 3, *P = 0.0051).

C  Effects of C$_{18}$-pyr-cer (10 μM, 1 h) on Drp1-MFF interaction in the absence/presence of E2F5 knockdown using shRNA (versus Scr-shRNA) were measured by immunoprecipitation followed by Western blotting using anti-Drp1 and anti-MFF antibodies (right panel). Equal immunoprecipitation of Drp1 or MFF was confirmed by Western blotting (left panel, input). Blots represent three independent studies. E2F5 knockdown was confirmed using qPCR (lower panel). Data are means ± SD from three independent experiments, analyzed by unpaired Student's *t*-test (*n* = 3, *P = 0.0041).

D  Effects of ectopic expression of E2F5 versus empty vector on Drp1-MFF or Drp1-MID49 (SMCR7) interaction in the presence/absence of C$_{18}$-pyr-cer (10 μM, 2 h) were measured by immunoprecipitation/Western blotting (right panels). Equal immunoprecipitation of Drp1, SMCR7 or MFF was confirmed by Western blotting (left panel, input). Ectopic expression of E2F5 was confirmed using qPCR (lower panel). Data are means ± SD from three independent experiments, analyzed by unpaired Student's *t*-test (*n* = 3, *P = 0.005).

E  Effects of shRNA-mediated E2F5 knockdown on Drp1 localization to mitochondria in the absence/presence of C$_{18}$-pyr-cer (20 μM, 1.5 h) were assessed in whole-cell lysates (UM-SCC-47) versus mitochondria-enriched fractions using Western blotting. Actin and Tom20 were used as controls for whole-cell and mitochondria-enriched fractions, respectively.

F  Effects of transient reconstitution of E2F5$^{WT}$ or E2F5$^{\Delta 84-177}$ proteins in UM-SCC-22A cells, which were stably transfected with E2F5-shRNA, on Drp1 abundance, were measured by Western blotting using anti-Drp1 antibody, in whole-cell lysates versus mitochondria-enriched fractions in the presence/absence of C$_{18}$-pyr-cer (20 μM, 1.5 h). Actin and Tom20 were used as controls for whole-cell and mitochondria-enriched fractions, respectively.

Data information: In all Western blot panels, images are representative of three independent experiments.
Source data are available online for this figure.

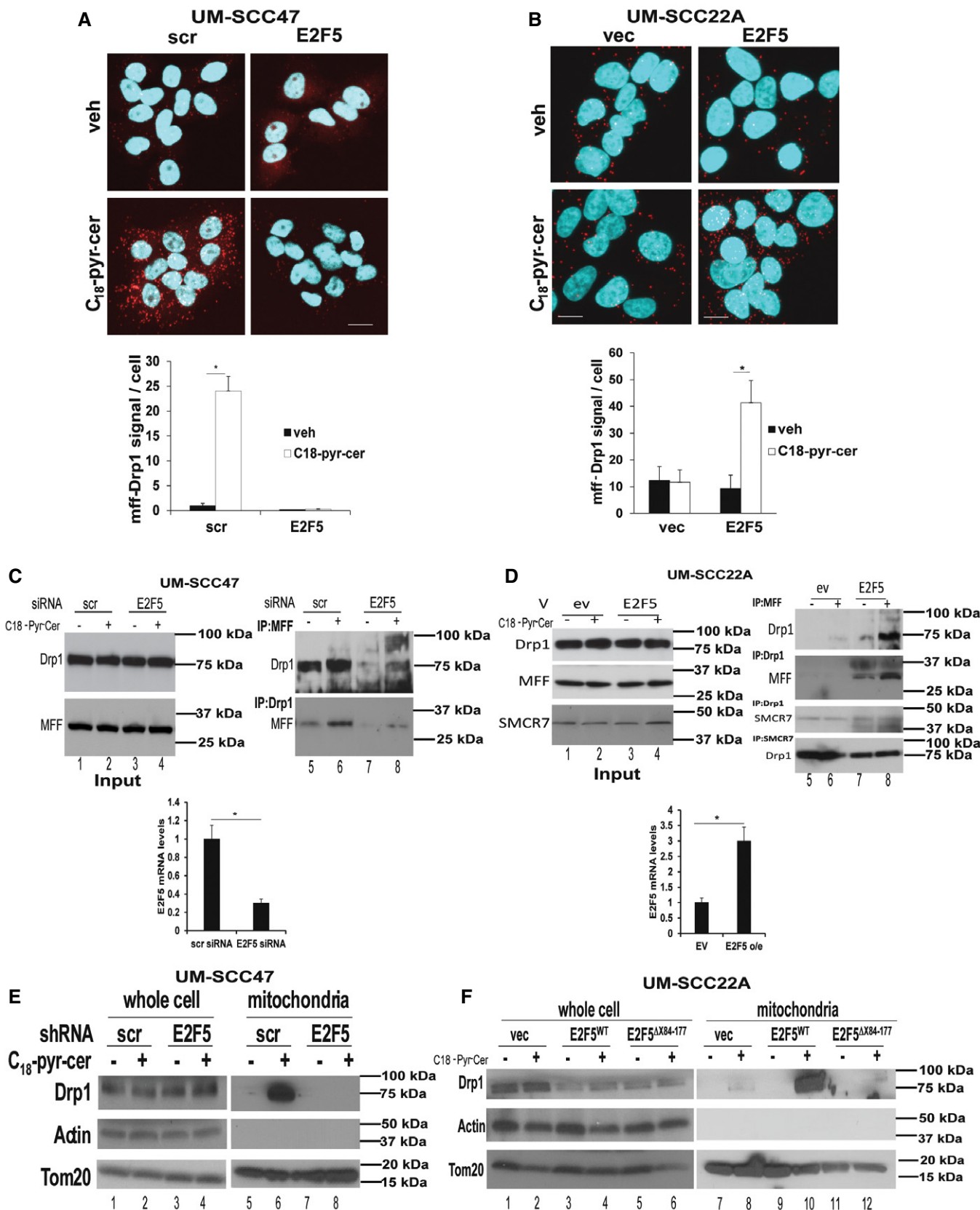

Figure 9.

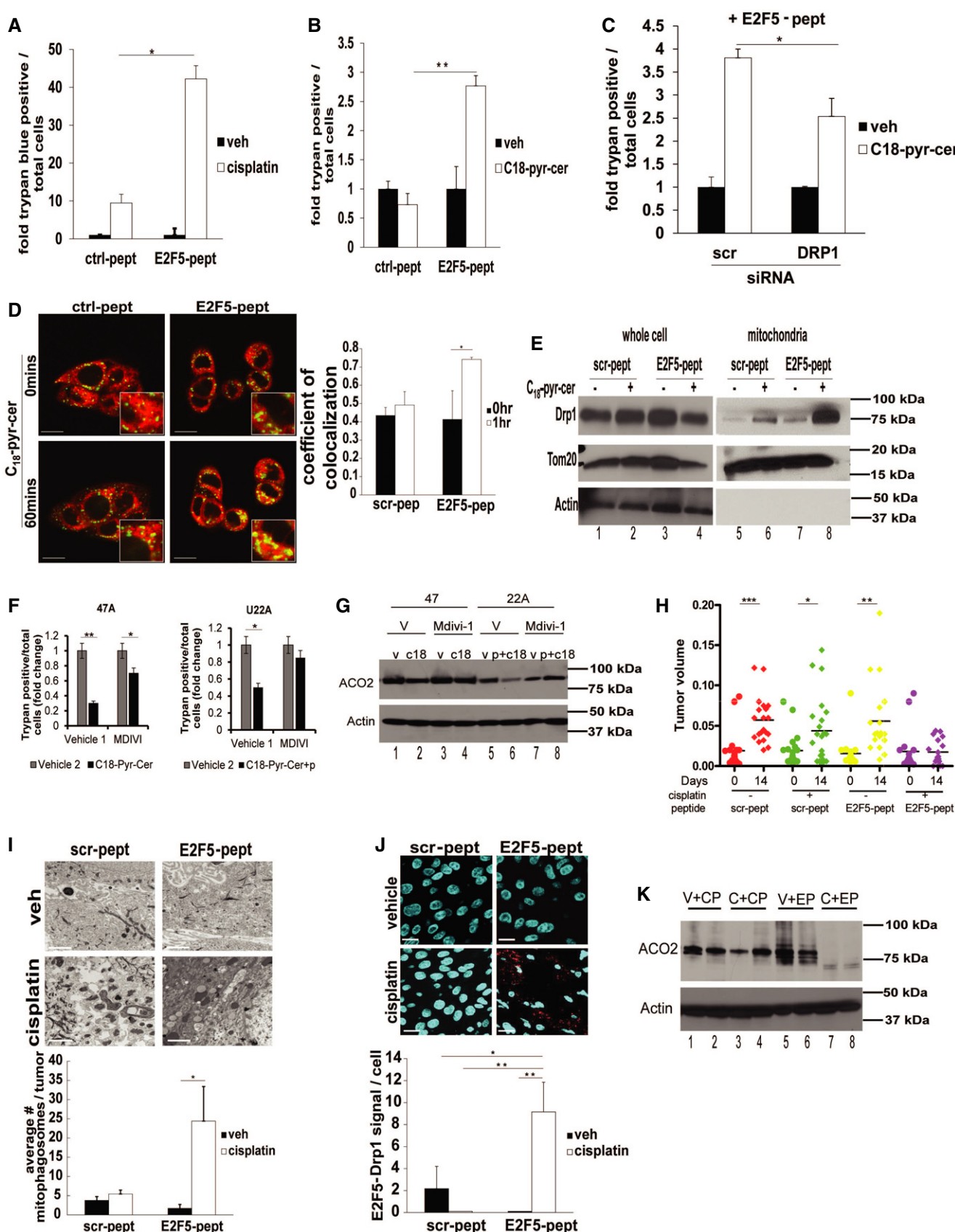

**Figure 10.**

◄ **Figure 10. Reconstitution of Drp1-E2F5 binding using an E2F5 peptide mimetic enhances ceramide-mediated lethal mitophagy in HPV(−) HNSCC cells and tumor xenografts.**

A, B UM-SCC2A cells were treated with a peptide designed to mimic the putative Drp1-binding region of E2F5 (E2F5-pept) or scrambled control peptide (Scr-pept) (5 μM), and their effects on cell death in response to cisplatin (40 μM, 48 h) (A) or $C_{18}$-pyr-cer (20 μM, 48 h) (B) were measured by trypan blue exclusion assay. Data are means ± SD from three independent experiments, analyzed by two-way ANOVA ($n = 3$, *$P = 0.0102$, **$P = 5.3 \times 10^{-5}$).

C Effects of shRNA-mediated knockdown of Drp1 on UM-SCC-22A cell death in the presence of E2F5-pept with/without $C_{18}$-pyr-cer were measured by trypan blue exclusion assay. Data are means ± SD from three independent experiments, analyzed by two-way ANOVA ($n = 3$, *$P = 0.0002$).

D Live cell imaging in UM-SCC-22A cells treated with/without $C_{18}$-pyr-cer was performed to measure the effects of Scr-pept (control) versus E2F5-pept as above using live cell imaging confocal micrographs of UM-SCC-22A cells stained with LTR and MTG. Images were quantified using ImageJ. Data are means ± SD from three independent experiments, analyzed by unpaired Student's *t*-test (*$P = 0.047$). Scale bars represent 100 μm.

E Drp1 localization to mitochondria in response to $C_{18}$-pyr-cer (20 μM, 1.5 h) in the presence of Scr-pept versus E2F5-pept (5 μM) was measured by Western blotting using whole-cell (UM-SCC-22A) lysates versus mitochondria-enriched fractions. Actin and Tom20 were used as controls for whole-cell and mitochondria-enriched fractions, respectively. In all Western blot panels, images are representative of three independent experiments.

F Effects of Drp1 inhibition using Mdivi (20 μM, 24 h pretreatment) on cell death in response to $C_{18}$-pyr-cer (10 μM, 24 h) alone in UM-SCC-47 (left panel) or with E2F5-peptide mimetic (10 μM) in UM-SCC-22A (right panel) cells were measured by trypan blue exclusion assay. Vehicle-treated cells were used as controls. Data are means ± SD from three independent experiments, analyzed by unpaired Student's *t*-test ($n = 3$, left panel *$P = 0.0378$, **$P = 0.0003$; right panel *$P = 0.0028$).

G Effects of Drp1 inhibition using Mdivi (20 μM, 24 h pretreatment) on ACO2 expression in response to $C_{18}$-pyr-cer (10 μM, 2 h) alone in UM-SCC-47 or with E2F5-peptide mimetic (10 μM) in UM-SCC-22A cells were measured by Western blotting using anti-ACO2 antibody. Vehicle-treated cells were used as controls. Actin was used as a loading control. Data represent three independent studies.

H Xenograft tumors generated from UM-SCC22A cells ($7.5 \times 10^4$ cells/100 μl PBS) in each flank of SCID mice were treated with cisplatin (3.5 mg/kg) or DMSO (veh) in the presence of E2F5-pept or Scr-pept (3.76 μg) for 14 days (every 3 days), and tumor volumes were measured using calipers at days 0 and 14 ($n = 5$–8 mice/group, and *$P = 0.045$, **$P = 0.0022$, ***$P = 0.0003$ by unpaired Student's *t*-test).

I Effects of E2F5-pept versus Scr-pept on mitophagy induction (by direct counting of mitophagosomes in TEM micrographs) in UM-SCC-22A xenograft-derived tumors isolated from SCID mice treated with Scr-peptide/vehicle, Scr-pept/cisplatin, E2F5-pept/vehicle and E2F5-pept/cisplatin were measured by TEM. Data are means ± SD from three independent experiments, analyzed by Student's *t*-test ($n = 5$–8, *$P = 0.0052$).

J Association between Drp1 and E2F5 was determined by PLA in tumor tissue sections isolated from SCID mice treated with Scr-pept/vehicle or Scr-pept/cisplatin versus E2F5-pept/vehicle or E2F5-pept/cisplatin (as in H) by PLA using labeled anti-Drp1 and anti-E2F5 antibodies. Data are means ± SD from 6 to 8 tumor tissue sections, analyzed by unpaired Student's *t*-test (*$P = 0.0068$, **$P = 0.0011$). Scale bars represent 100 μm.

K ACO2 expression in UM-SC-22A xenograft-derived tumor tissues isolated from SCID mice as in (H) treated with Scr-pept/vehicle (V+CP) or Scr-pept/cisplatin (C+CP) versus E2F5-pept/vehicle (V+EP) or E2F5-pept/cisplatin (C+EP) ($n = 2$ tissues/sample) was measured by Western blotting using anti-ACO2 antibody. Actin was used as a loading control.

Source data are available online for this figure.

degradation (Scheffner *et al*, 1990). E7 binds RB and inhibits cell cycle restriction by activation of E2F transcription factors (Helt *et al*, 2002). There are eight reported members of E2F family (Lee *et al*, 2011) with diverse roles in cell cycle regulation and development (Chen *et al*, 2009; Ambrus *et al*, 2013). Among those, RB inhibits E2F, which, upon E7-RB binding, is relieved/activated and plays various nuclear and cytosolic functions. Liberated E2F from RB induces cell cycle exit and differentiation (Hijmans *et al*, 1995; Itoh *et al*, 1995; Apostolova *et al*, 2002), as opposed to cell cycle progression by E2F1. Specific roles of HPV16-E7/RB/E2F signaling in HNSCC cell death for improved therapeutic response are largely unknown. However, involvement of infected cells' impaired ability to repair DNA damage (Bol & Grégoire, 2014) or priming of the immune system (Rieckmann *et al*, 2013), as seen in response to radiation therapy, were reported in improved response of HPV(+) HNSCC patients to therapy previously.

Cisplatin is one of the most commonly used chemotherapeutics for HNSCC, where it forms DNA adducts, inducing apoptosis (Ziemann *et al*, 2015). It is also known that cisplatin induces mitochondrial stress, consistent with the induction of CerS1/ceramide accumulation in mitochondria, enhancing lethal mitophagy in response to HPV-E7 signaling. Although ceramide transport from ER to Golgi is mediated by CERT (Hanada *et al*, 2003), the transport mechanisms for ceramide and/or CerS1 enzyme from the ER to mitochondria to induce ceramide-dependent mitophagy in response to cellular stress in HPV(+) cells remain unknown.

Induction of lethal mitophagy has not previously been evaluated in HPV-associated cancers; however, increased autophagy has been observed in cervical cancer cells in response to the

chemotherapeutic drugs paclitaxel or resveratrol (García-Zepeda *et al*, 2013). Our data are consistent with these previous studies, which support that ceramide-dependent lethal mitophagy/autophagy might play a key role in this process. Importantly, mitochondrial dysfunction can result in Rb hypophosphorylation and therefore inactivation of E2F (Gemin *et al*, 2005). These studies indicate that a feedback mechanism may exist between mitophagy and Rb/E2F signaling in HPV-associated cancers. Consistent with our data, HPV-E7 might render the hypophosphorylated RB inactive, allowing E2F5 to associate with Drp1 and promote trafficking to mitochondria, enhancing lethal mitophagy. This scaffolding role of E2F5 for Drp1 activation to induce lethal mitophagy without a need for co-expression of the dimerization partner protein (DP), which is required for exogenous E2F to be active as a transcription factor (Tao *et al*, 1997; Chang *et al*, 2004; Sadasivam & DeCaprio, 2013), has not been described previously. Association of Drp1 with other proteins involved in mitochondrial fission leading to induction of mitophagy, such as MFF and FUNDC1, was demonstrated previously (Wu *et al*, 2016; Zhang & Lin, 2016). However, induction of ceramide-dependent mitophagy via activation of E2F5-Drp1 complex via targeting/inhibition of RB by HPV-E7 is novel.

Overall, our data suggest that activation of ceramide signaling in HPV-associated HNSCC plays a key role in stress-mediated cell death and tumor suppression. The present research provides important mechanistic implications for improving therapies in non-HPV-associated HNSCC in future studies. This is supported by our E2F5 peptide mimetic, which can recapitulate the inhibition of RB by HPV-E7 and resultant increased translocation of Drp1 to the mitochondria, mitochondrial fission, and lethal mitophagy. Thus,

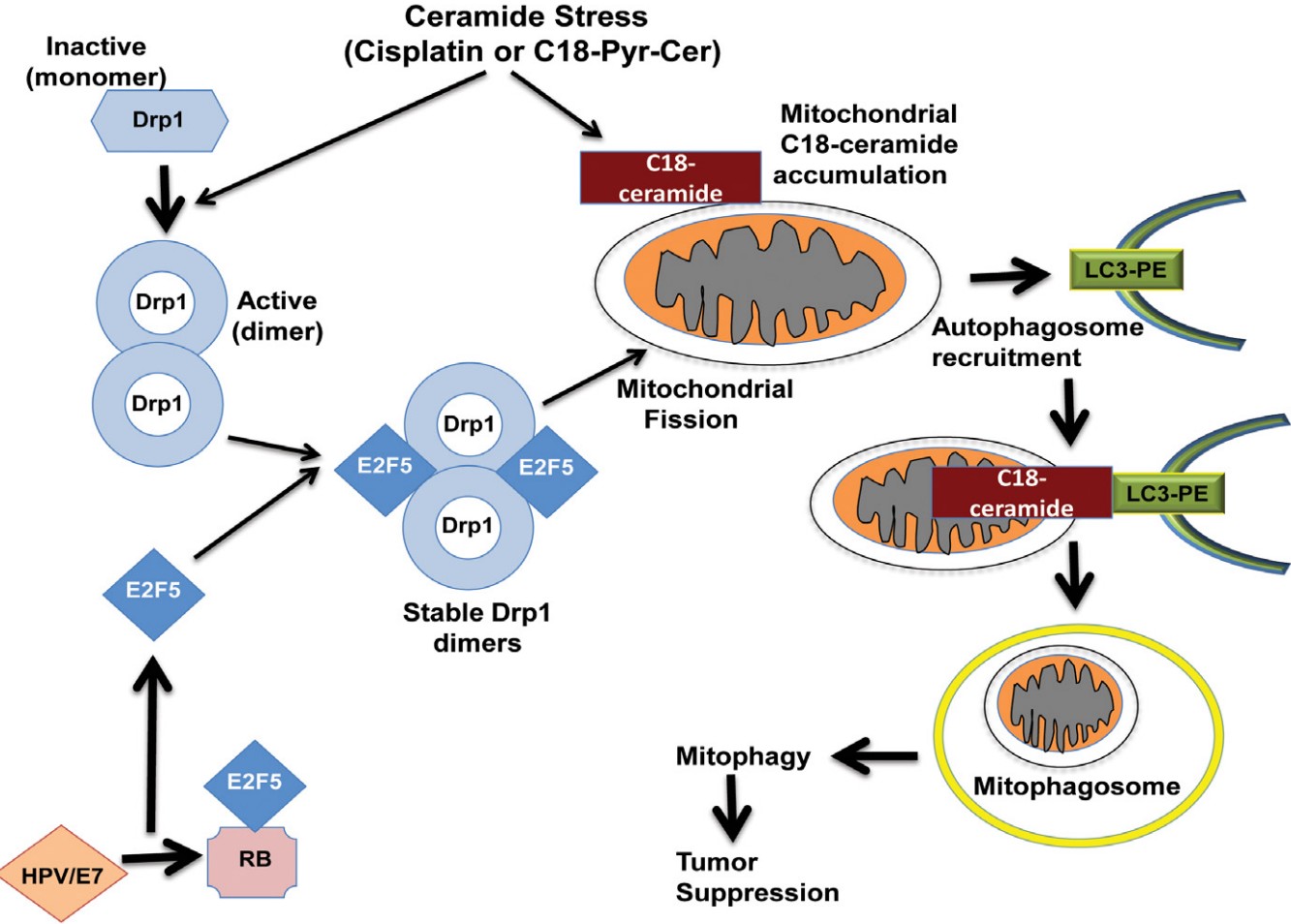

**Figure 11.  Graphical summary.**
Our data suggest that inhibition of retinoblastoma protein (RB) by the HPV-E7 oncoprotein relieves E2F5, which then associates with Drp1 as a scaffolding protein, resulting in Drp1 activation and/or oligomerization, leading to increased mitochondrial localization of Drp1, resulting in mitochondrial fission, ceramide-dependent lethal mitophagy, and tumor suppression.

development of new strategies to improve therapeutic outcome of patients with HPV(−) HNSCC using our mechanistic data might have significant impact to improve chemotherapy-mediated tumor suppression by at least in part inducing E2F5-Drp1-ceramide-dependent lethal mitophagy without the oncogenic HPV infection.

# Materials and Methods

### Reagents

$C_{18}$-pyridium-ceramide was synthesized at the synthetic Lipidomics Core at the Medical University of South Carolina (MUSC). Cisplatin was purchased from Sigma. Treatments were performed using 40 μM cisplatin in DMSO, 20 μM $C_{18}$-pyr-cer in EtOH for 1–4 h for mitophagy detection, or corresponding amount of vehicle control. Peptides were synthesized by LifeTein, Inc. Peptides contained C-terminal amidation.

E2F5-pept: Biotin-RRRRRRRR-ELDQQKLWLQQSIKNVMDDSINN RFSYVTHED.

Scr-pep: Biotin-RRRRRRRR-LILFVIKLHQDVNDMRNSNQDQTQSE DRESKWY.

### Antibodies

Tom20 (sc-17764, Santa Cruz Biotechnology), CERS1 (sc-65096, Santa Cruz Biotechnology); E2F5 (E-19, sc-999, Santa Cruz Biotechnology), HPV16 E6 (N-17, sc-1584, Santa Cruz Biotechnology), HPV16 E7 (NM2, sc-65711, Santa Cruz Biotechnology); lamin B (C-20: sc-6216, Santa Cruz Biotechnology); clathrin (TD.1, sc-12734, Santa Cruz Biotechnology); ceramide (MID15B4, Enzo Life Sciences); actin (A2066, Sigma, Inc.), Drp1 (BD# 611738, Clone 22/ Drp1, BD Biosciences), ACO2 (#6922, Cell Signaling Technology); MFF (ab81127, Abcam); LC3 (D11 XP® mAb #3868, Cell Signaling Technology); ATG5 (#2630, Cell Signaling Technology); p53 (7F5, Rabbit mAb #2527, Cell Signaling Technology); pRb (4H1, Mouse mAb #9309, Cell Signaling Technology), SQSTM1/p62 (#5114, Cell Signaling Technology); SQSTM1/p62 (#5114, Cell Signaling Technology), SMCR7/MIEF2/MID49 (#PA5-46624, Invitrogen).

## Cell lines and culture conditions

HPV(+) cell lines were provided by Drs. Tom Carey, University of Michigan (UM-SCC-47) and Susan Gollin, University of Pittsburgh Cancer Institute (UPI-SCC-90). UM-SCC-22A, UM-SCC-1A, and UM-SCC-47 were grown in DMEM (Corning) with 10% FBS (Atlanta Biologicals) and 1% Penicillin and streptomycin (Cellgro) at 37°C with 5% $CO_2$. UPI-SCC-90 were grown in DMEM with 10% FBS (Atlanta Biologicals), 2 mM L-glutamine, 1× non-essential amino acids solution, and 500 μg/ml gentamicin (Gibco). All cell lines were tested for mycoplasma contamination. These cell lines were authenticated via short tandem repeat (STR) profiling (Power-Plex16HS) by Genetica DNA Laboratories (Burlington, NC) in November 2016.

## Stable shRNA-mediated knockdown of E2F5

pLKO.1 plasmids expressing shRNA to E2F5 or scrambled control (MUSC shRNA Shared Technology Resource) were co-transfected with pCMV-psPAX2 and pMD2 plasmids in Plat A cells using the viral transduction protocol as described by the RNA interference (RNAi) Consortium. Viral supernatants were added to UM-SCC47 cells, and selection was performed using puromycin (1 μg/ml) for 14 days.

## Cell transfections

Plasmids containing target gene, shRNA, or empty vector were transfected into cells using effectene transfection reagent (QIAGEN) following the manufacturer's instructions, followed by PBS rinse 6 h after transfection. CerS6, Drp1, p53, Rb, and E2F1, 4, and 5 shRNAs were from the MUSC shRNA Shared Technology Resource. siRNA transfections were performed using DharmaFECT™ (ThermoScientific, Dharmacon). SiRNAs used: E6/E7 (ThermoScientific, custom sequence AGGAGGAUGAAAUAGAUGGUU); CerS1, ATG5, LC3B (Thermo Scientific, Dharmacon); or non-targeting scrambled siRNA (Qiagen).

## Trypan blue exclusion assay

Cells were seeded in 6-well plates and allowed to adhere for 20 h. After treatments (for 24 h), medium containing dead cells was pelleted with trypsinized cells, then re-suspended in 1× PBS then counted in a hemocytometer after addition of trypan blue dye (Sigma-Aldrich) at a 1:10 dilution.

## $IC_{50}$ determination by MTT assay

Cells were plated in 96-well plates and allowed to adhere for 20 h, then treated with the indicated concentrations of drug. After 48-h treatment, the MTT assay (ATCC) was performed as described previously (Sentelle *et al*, 2012).

## Immunoblotting

Cells were lysed in RIPA buffer plus protease inhibitor cocktail on ice for 15 min then centrifuged. 30 μg of protein from the supernatant was run on Criterion™ TGX™ Precast Gels (Bio-Rad).

## Quantitative RT–PCR

RNA was extracted from cell pellets using RNeasy kit (Qiagen) per the manufacturer's instructions. cDNA was generated using equal amounts of RNA from each sample and iScript cDNA synthesis kit (Bio-Rad) per manufacturer's instructions. Reactions were carried out using SsoFast probes mix (Bio-Rad) and TaqMan primer probes (ThermoFisher Scientific) in a StepOne Plus qPCR cycler as described by the manufacturer.

## Site-directed mutagenesis

$E2F5^{\Delta X84-177}$ mutant was generated using Q5 site-directed mutagenesis kit (New England Biolabs) per manufacturer's instructions. Primers were designed using NEBaseChanger (New England Biolabs). Forward: GAG GTG GAG GTC TAG ATC ACC AAT GTC TTA GAG GG, Reverse: CAG TGT GGT GGA ATT CTA TGT ATC ACC ATG AAA GC.

## Measurements of $O_2$ consumption rate (OCR)

A Seahorse Bioscience XF24 instrument was used to measure the rate of change of dissolved $O_2$ in medium immediately surrounding UMSSC47 cells cultured in customized XF96 96-well plates overnight at 37°C. Cells were treated with 20 μM C18-pyr-ceramide or vehicle for 2 h at 37°C. Measurements were performed using a cartridge where 96 optical fluorescent $O_2$ sensors were configured as individual well "plungers". For measurements of rates, the plungers were descended into the wells, forming a chamber that entraps the cells. $O_2$ concentration was measured over 1 min. The rates of $O_2$ consumption were obtained from the slopes of concentration changes versus time. For preparation of the cell plate for assay with the XF96 instrument, 1 ml of Krebs-Henseleit buffer lacking bicarbonate (111 mM NaCl, 4.7 mM KCl, 2.0 mM $MgSO_4$, 1.2 mM $Na_2HPO_4$, 0.24 mM $MgCl_2$, 2.5 mM glucose, 0.5 mM carnitine, and 100 nM insulin) at 37°C was added to each well containing $1.8 \times 10^4$ cells. Various inhibitors were introduced by automatic injectors followed by brief (15 s) mixing as follows: port A injection, media; port B injection, 1 μM oligomycin; port C injection, 1 μM FCCP, carbonyl cyanide-4-(trifluoromethoxy) phenylhydrazone; port D injection, 1 μM antimycin A, and 2 μM rotenone (Beeson *et al*, 2010).

## Molecular modeling of protein–protein interactions

Phyre2 was used to predict secondary structure of E2F5 based on the sequence in GenBank (NP_001942.2). The generated PDB file was input along with the PDB for Drp1(4BEJ) from RSCB (www.rscb.org) into ZDOCK Server (http://zdock.umassmed.edu/). The top model was then used to predict the sites of association between E2F5 and Drp1.

## Measurement of cellular respiration using Seahorse XF analyzer

Cells were plated in a Seahorse Biosciences 96-well plate and allowed to adhere for 20 h. They were then treated with 20 μM $C_{18}$-pyr-cer or equivalent amount of vehicle (EtOH) for 2 h, and then, oxygen consumption rate was measured using a Seahorse XF96 (Seahorse Biosciences) as described by the manufacturer (Sentelle *et al*, 2012).

## Proximity ligation assay

Proximity ligation assays were performed using Duolink *in situ* red kit (Sigma) per manufacturer's instructions, then analyzed as described (Panneer Selvam *et al*, 2015).

## Immunofluorescence

Cells were plated on glass coverslips in 6-well plates and allowed to adhere for 20 h. Treatment was performed with 40 μM cisplatin or equivalent amount of DMSO for 8 h, or 5 μM peptide for 2 h. Fixation was in 4% paraformaldehyde, followed by permeabilization with 0.1% Triton X-100, and blocking in 1% BSA in PBS. Samples were incubated at 4°C overnight with primary antibodies in blocking solution. Tom20 (SCBT) 1:200, Lass1/CerS1 (SCBT) 1:50, ceramide (Enzo Life Sciences) 1:100, and biotin (SCBT) 1:200. Immunofluorescent-conjugated secondary antibodies (AlexaFluor 488 or 594, Jackson Immuno) were added at 1:500 for 1 h. Coverslips were then mounted onto glass slides with ProLong® Gold Antifade Mountant (Molecular Probes).

## Laser scanning confocal microscopy

For live cell imaging, cultured cells were incubated with 500 nM of MitoTracker far red and 500 nM LysoTracker green in DMSO for 30 min at 37°C. Cells were treated with 20 μM $C_{18}$-pyr-cer or 40 μM cisplatin and kept in an incubator with 5% $CO_2$ at 37°C during imaging. An Olympus FV10i confocal microscope was used for imaging. 543- and 488-nm channels were used for visualizing red or green fluorescence, respectively, with pinholes set to 1.0 Airy units. At least three random fields were imaged for each sample (Sentelle *et al*, 2012).

## Ultra-structure analysis using transmission electron microscopy

Cells were washed with 1× PBS then fixed in 2% glutaraldehyde (w/v) in 0.1 M cacodylate. After post-fixation in 2% (v/v) osmium tetroxide, specimens were embedded in Epon 812, and sections were cut orthogonally to the cell monolayer with a diamond knife. Thin sections were visualized in a JEOL 1010 transmission electron microscope (Saddoughi *et al*, 2013).

## Cell fractionation

Cells were treated with 40 μM cisplatin or vehicle for 8 h, 20 μM $C_{18}$-pyr-cer or vehicle and/or 5 μM scr-pept or E2F5-pept for 1.5 h. Mitochondria isolation kit (ThermoFisher Scientific) was used as described by the manufacturer.

## Co-immunoprecipitation

Cells were lysed in 500 μl Pierce™ IP lysis/wash buffer (Thermo-Fisher) with protease inhibitor cocktail (Sigma) on ice for 15 min. 350 μg of protein was used with SureBeads™ Protein A Magnetic Beads (Bio-Rad) per manufacturer's instructions, using 10 μg antibody or corresponding normal IgG control (SCBT).

## Image quantification

Images were quantified by ImageJ. Duolink ImageTool software was used for quantification of PLA signals (Panneer Selvam *et al*, 2015).

## *In vivo* studies

Severe combined immunodeficient (SCID) mice were purchased from Jackson Laboratories. Age- and sex-matched mice were used. All animal protocols were approved by the Institutional Animal Care and Use Committee at the Medical University of South Carolina. UM-SCC22A or UM-SCC47 cells (75,000) were implanted into the flanks of SCID mice ($n$ =5–8 mice). When the tumors were palpable, the mice were treated every 3 days with 3.5 mg/kg cisplatin, 20 mg/kg $C_{18}$-pyr-cer, or corresponding amount of vehicle control and/or 3.76 μg E2F5-peptide or scrambled control peptide. Tumor volume was measured using calipers. At the end of the 14-day treatment, the mice were euthanized and tumor tissues were collected (Sentelle *et al*, 2012; Saddoughi *et al*, 2013).

## Statistical analyses

Data were reported as mean ± standard error. Mean values were compared using the Student's *t*-test or ANOVA, and $P < 0.05$ was considered statistically significant (Saddoughi *et al*, 2013). In animal studies, the group sizes were calculated based on 80% confidence intervals. The comparison of two groups was based on the assumption of normal distribution and was carried out with the two-sample *t*-test. For the comparison of several groups, a variance analysis (ANOVA) was carried out under normal distribution assumption.

**Expanded View** for this article is available online.

## Acknowledgements
We thank Lipidomics Shared Resource Facility, Medical University of South Carolina (MUSC); Cell and Molecular Imaging Core; and the shRNA Shared Technology Resource (MUSC) for their valuable services. We also thank Drs. Dennis Watson, Ashley Cowart, Natalie Sutkowski, and Steven Rosenzweig (MUSC) for their valuable input and feedback. We thank Drs. Gyda and Craig Beeson (MUSC) for assistance with the Seahorse Biosciences analyzer, and Dr. Elizabeth Garrett-Mayer and Mr. Kent Armeson (MUSC, Hollings Cancer Center) for their help for statistical analyses. We thank Drs. Tom Carey (University of Michigan) and Susan Gollin (University of Pittsburgh Cancer Institute) for HPV-positive cell lines (UM-SCC47 and UPI-SCC90); Dr. Denise Galloway (Fred Hutchinson Cancer Research Center) for providing HPV16-E6 and E7 plasmids; Dr. Nick Dyson (Massachusetts General Hospital), for providing the RB10 plasmid; and Dr. Lina Dagnino (University of Alberta) for providing the E2F5 plasmid. This work was supported by research funding from the National Institutes of Health (NIH) (DE16572, CA88932, CA173687 and CA203628-01 (P01) to B.O.; DE016572-07S1 and 5T32-DE017551 to R.J.T.). Institutional resources at the Medical University of South Carolina were supported by NIH support C06 RR15455 and P30 CA138313 grants (to Hollings Cancer Center).

## Author contributions
RJT designed and performed experiments, analyzed data, prepared the figures, and helped write the manuscript; NO, SPS, SGV, MD, RNN, RD, JK, and KDB performed experiments; ZMS synthesized ceramide analogues; BO designed experiments, analyzed data, and wrote the manuscript.

**The paper explained**

**Problem**

Molecular mechanisms by which human papillomavirus (HPV) induces stress-mediated head and neck cancer cell death are largely unknown.

**Results**

We demonstrate here that inhibition of retinoblastoma protein (RB) by the HPV-E7 oncoprotein relieves E2F5, which then associates with Drp1 as a scaffolding protein, resulting in Drp1-mediated mitochondrial fission, induction of ceramide-dependent lethal mitophagy, and tumor suppression.

**Impact**

The discovery of the mechanism by which HPV-E7 induces stress-mediated cell death, at least in part, by ceramide-dependent lethal mitophagy, has important implications for identification of novel targets to improve treatments in various cancers, such as cervical and HNSCC, without pathogenic HPV infection.

## Conflict of interest

The authors declare that they have no conflict of interest.

## For more information

Lipidomics: http://www.hollingscancercenter.org/research/shared-resources/lipidomics/index.html

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
