## [Review Process File · EMBO Molecular Medicine]

HPV/E7 induces chemotherapy-mediated tumor suppression by ceramide-dependent mitophagy

Raquela J. Thomas, Natalia Oleinik, Shanmugam Panneer Selvam, Silvia G. Vaena, Mohammed Dany, Rose N. Nganga, Ryan Depalma, Kyla D. Baron, Jisun Kim, Zdzislaw M. Szulc, Besim Ogretmen

Corresponding author: Besim Ogretmen, Medical University of South Carolina

Review timeline:

Submission date:	19 September 2016
Editorial Decision:	04 November 2016
Revision received:	30 March 2017
Editorial Decision:	21 April 2017
Revision received:	01 May 2017
Accepted:	04 May 2017

Transaction Report:

Editor: Roberto Buccione

1st Editorial Decision

04 November 2016

Thank you for the submission of your manuscript to EMBO Molecular Medicine. We are very sorry that it has taken so long to get back to you on your manuscript.

In this case we experienced unusual difficulties in securing three willing and appropriate reviewers. As a further delay cannot be justified I have decided to proceed based on the two available consistent evaluations.

As you will see, both reviewers clearly find that there is promise in your work but note, with varying degrees of concern, that you have not succeeded in providing sufficient support for your main conclusions, including the mitophagy link, not taking into sufficient consideration the broader effects of ceramide, and the link between putative ceramide-induced mitophagy and tumour suppression. They also note several inconsistencies, the need for additional controls, and a certain degree of oversimplification, poor image presentation, inadequate statistical analysis and other issues.

While publication of the paper cannot be considered at this stage, we would be willing to consider an extensively revised submission, with the understanding that both Reviewers' concerns must be fully addressed including with additional experimental data where appropriate and that acceptance of the manuscript will entail a second round of review.

Please note that it is EMBO Molecular Medicine policy to allow a single round of revision only and that, therefore, acceptance or rejection of the manuscript will depend on the completeness of your responses included in the next, final version of the manuscript.

Should you find that the requested revisions are not feasible within the constraints outlined here and choose, therefore, to submit your paper elsewhere, we would welcome a message to this effect.

As you know, EMBO Molecular Medicine has a "scooping protection" policy, whereby similar findings that are published by others during review or revision are not a criterion for rejection. However, I do ask you to get in touch with us after three months if you have not completed your revision, to update us on the status. Please also contact us as soon as possible if similar work is published elsewhere.

Please note that EMBO Molecular Medicine now requires a complete author checklist (<http://embomolmed.embopress.org/authorguide#editorial3>) to be submitted with all revised manuscripts. Provision of the author checklist is mandatory at revision stage; The checklist is designed to enhance and standardize reporting of key information in research papers and to support reanalysis and repetition of experiments by the community. The list covers key information for figure panels and captions and focuses on statistics, the reporting of reagents, animal models and human subject-derived data, as well as guidance to optimise data accessibility. The Author checklist will be published alongside the paper, in case of acceptance, within the transparent review process file

We now mandate that all corresponding authors list an ORCID digital identifier. You may do so though our web platform upon submission and the procedure takes <90 seconds to complete. We also encourage co-authors to supply an ORCID identifier, which will be linked to their name for unambiguous name identification.

I look forward to seeing a revised form of your manuscript in due time.

***** Reviewer's comments *****

Referee #1 (Remarks):

The manuscript entitled HPV/E7 induces chemotherapy-mediated tumor suppression by ceramide-mediated mitophagy by R.J. Thomas describes a link between the expression of the human papillomavirus protein E7 in head and neck squamous carcinoma cells and the induction of mitophagy. The mechanism seems to be mediated by an E7-mediated inhibition of RB, which results in an association of E2F5 and Drp1, Drp1 activation and mitochondrial association finally resulting in mitophagy. E7 thereby enhances ceramide-triggered mitophagy. The manuscript is interesting, but several major issues remain:

1. While the link between E7, E2F5 and Drp1 is clearly and convincingly provided, the further link to mitophagy is based on correlation. The manuscript does not provide proof that ceramide-induced mitophagy causes tumor suppression. The data only provide a correlation. Therefore, it is absolutely critical to provide clear experimental evidence that in the present model ceramide-induced mitophagy is really mediating tumor suppression or is involved in the regulation of tumor progression. The data link the described pathways to mitophagy, but they do not prove that mitophagy is the mechanisms how E7 may sensitize and ceramide kill tumor cells.
2. Ceramide has many functions, among them the induction of cell death. However, it is too simple to describe ceramide as an exclusive pro-cell death molecular. In many systems it is a stress mediator, in some systems it even promotes cell growth and stimulation. Further, it is very important to discriminate the compartments in which ceramide has been generated and accordingly the enzymes or pathways that released or synthesized ceramide from pre-cursors. Several of the statements in the manuscript are too general.
3. The data that link CerS1 to mitophagy are convincing and provide re-constitution experiments. They are done in one cell line only and need to be repeated in second cell line.

4. Fig. 2C shows a minimal difference between vector + cisplatin (red curve) and E7 and cisplatin (orange) only at doses between 6.25 and 25 nM cisplatin. The difference is biologically irrelevant. Further, E6-transfected cells also almost do not differ from the E7-transfected cells. At 50 nanomolar all cells are dead, at 25 nM approximately 30% vs 20% (E7 transfected). Such a small difference does not allow the very general conclusion in the present manuscript.
5. Knock-down of LC3B and Atg5 prevents C18-pyr-cer mediated mitophagy, but this knock-down will also prevent other forms of autophagy and does not prove that mitophagy is the mechanism by which E7 facilitates killing of the tumor cells.
6. It is difficult to understand how a dose dependent difference in mitophagy in the different cells treated with ceramide analogs fits to the complete absence of mitophagy in E7-negative cells.
7. Several studies show trypan blue positive cells between 6 and 10% at maximum. This is again irrelevant in a tumor cell line.
8. Fig. 1A, Fig. 2D and many other fluorescence microscopy images are overexposed and some of the co-localizations might be simply indicated by the overexposure. Please also quantify the overlap of the two signals.

Referee #2 (Comments on Novelty/Model System):

1. The authors are using a number of excellent techniques with high quality throughout the manuscript.
2. Novelty and Medical Impact: No one has determined why HPV status correlates to chemotherapy outcomes, and this paper demonstrated why and orients the signaling pathway.
3. Model systems are standard and appropriate.

Referee #2 (Remarks):

This review is for the manuscript entitled "HPV/E7 induces chemotherapy-mediated tumor suppression by ceramide-dependent mitophagy," authored by Raquela J. Thomas et al. and submitted to EMBO Journal.

Results: HPV(+) HNSCC responds to cisplatin chemotherapy and ceramide treatment, whereas the response by HPV(-) HNSCC to these agents is attenuated. The authors show that HPV-mediated HNSCC cell death is CerS1/ceramide-dependent, and that this death occurs via a lethal mitophagy. The authors then demonstrate that this response occurs via the E7 HPV oncogene, and that this factor targets an Rb/E2F5/Drp1/MFF axis in the mitophagic response. Finally, the authors show that an E2F5 peptide may have some efficacy in the treatment of HPV(-) HNSCC in a murine model.

Conclusions: The authors conclude that activation of ceramide signaling in HPV-associated HNSCC plays a key role in the lethal mitophagy observed in this model, and that these pathways, if activated exogenously, can be used to target HPV(-) HNSCC for cell death.

Review: This manuscript represents a full analysis of mitophagic pathways induced by ceramide in HPV(+) HNSCC and which can be leveraged to enhance cell death in HPV(-) HNSCC. This manuscript is potentially of great importance to the field, and most of the data are of high quality. However, there are a few deficiencies which are described below.

Major criticism 1: All mitophagy panels: Although colocalization of lysotracker and mitotracker may be an indication of mitophagic flux, this method is not a definitive one, and is especially problematic as, by eye, the C18-ceramide treatment seems to enhance the mitotracker red intensity in some instances. It is also unclear as to how mitochondrial staining is enhanced with mitophagic signaling, as one would predict that targeting mitochondria for autophagy would reduce mitochondria number/function. The authors should provide some corroborating data such as mitochondria dyes for total numbers versus function and simple westerns for mitochondria.

Major criticism 2: Figures 3-4: There is some concern that the ceramide treatment used to induce mitophagy is having a general metabolic effect, especially as the appropriate controls have not been followed. Although later figures alleviate these concerns somewhat, using control species would enhance enthusiasm for this manuscript. Indeed, Pyr-Cer is in no way specific on many levels for either just mitochondrial targeting or chain length specificity when used as an exogenous agent.

Moderate criticism 1: Figure 2D-2E: The timing of cisplatin treatment and mitophagy is unclear in these figures. The authors claim that mitophagy (which is predicted to translate into fewer mitochondria/cell, as the mitochondria flux through the autophagic pathway) is occurring in Fig. 2D. However, in Fig. 2E, the mitochondrial marker Tom20 is increasing with increased cisplatin treatment. Furthermore, the immunoblot quantification results seem not to match the image. Addressing these concerns would be valuable.

Moderate criticism 2: Throughout the manuscript, the authors have not used appropriate statistical tests. Student's t-test is only appropriate when there are only 2 experimental conditions. An ANOVA with appropriate post-hoc test should be used in most instances in this manuscript.

Moderate criticism 3: Figure 9A-B and Figure 10H: The MFF/Drp1 colocalization micrographs are small and difficult to interpret. A coIP (similar to Figure 7E) is highly recommended to take their place.

Moderate criticism 4: Throughout the manuscript: Although the respiration data are very nice, some data correlating respiration rate to total mitochondria per cell would be more informative as to mitochondrial performance.

Moderate criticism 5: Figure 3D: The data here demonstrate that ATG5 "knock-down" basally enhances mitochondria/lysosome colocalization. This is, at first glance, unexpected and should be addressed.

Minor criticism 1: Throughout the manuscript, there are multiple mis-labeling errors. For example, in the legend for Figure 1 A-B mentions image quantification, but a graph does not appear in the figure. Furthermore, Figure 2 legend title "HPV-E6 enhances chemotherapy-mediated..." should likely read "HPV-E7 enhances...". In Figure 1D, the asterisks are askew and in Figure 6A, the line above the statistically significantly different shRNAs is not centered correctly.

Minor criticism 2: Throughout the manuscript, panel labels are not centered and/or are small and thus difficult to read. In addition, some panels are labeled with the cell line used and others are not, and the labeling of many y-axes are inconsistent (for example, some read "trypan blue positive cells" and others read "fold trypan positive/total cells"). More consistent and larger labeling for clarity would be greatly appreciated. One should aspire to the Carmen Guidelines for Figure production especially for more high impact journals.

Minor criticism 3: The authors make a few odd choices as to figure panel organization. For example, the wonderful pathway figure which ties all the data together is relegated to the "expanded view" and not showcased in the main manuscript. As another example, some of the panels in Fig. 4 might be better added to Figure 3 to make the manuscript more easily interpreted.

Minor criticism 4: "Sea Horse" is not defined except in the methods section, which was confusing.

Minor criticism 5: Figure 1A: Are the wild-type MEFs larger than the mutant cerS MEFs? By eye, based on the micrographs shown, it would seem so.

Minor criticism 6: Figure 5A: Although the Rb "knock-down" shown originates on the same blot, a repeat run using seque

We thank the reviewers for their careful reviews and helpful comments. We have addressed the points raised by the reviewers in our revised manuscript as follows:

Reviewer 1:

We thank the reviewer for finding the manuscript interesting. Below is our point-by-point responses to the reviewer's comments:

1. While the link between E7, E2F5 and Drp1 is clearly and convincingly provided, the further link to mitophagy is based on correlation. The manuscript does not provide proof that ceramide-induced mitophagy causes tumor suppression. The data only provide a correlation. The data link the described pathways to mitophagy, but they do not prove that mitophagy is the mechanisms how E7 may sensitize and ceramide kill tumor cells.

Response: To address these important points, we have performed various studies. First, we have used molecular approach to prevent mitophagy by Drp1 knockdown using shRNA. This attenuated cisplatin and C18-ceramide-mediated HNSCC growth suppression (in the presence of E2F5-peptide) (Fig. 10C). Next, we have inhibited Drp1 activity using Mdivi1, a known pharmacologic inhibitor of Drp1, which also resulted in attenuation of ceramide-mediated mitophagy and prevented growth suppression Fig. 10F-G. We also provide data which support that E7 sensitizes the oral cancer cells, at least in part, via inducing mitophagy, as inhibition of mitophagy by shRNA-mediated ATG5 or LC3 knockdown, prevented growth suppression in HPV/E7+ HNSCC cells (Figs. 3C, 3D and 3E). Then, we also determined the effects of Drp1 knockdown using shRNAs on E7-mediated HNSCC growth suppression and mitophagy in response to cisplatin. These data also showed that inhibition of mitophagy by Atg5 or Drp1 knockdown prevented E7-mediated HNSCC growth inhibition in response to cisplatin (Fig. 2D and EV2A-B). It should also be noted that we have tried to repeat these studies in vivo, however, stable knockdown of Drp1 to prevent mitophagy in HNSCC cells expressing HPV-E7 have been challenging for long-term xenograft growth and treatment studies in SCID mice, which take around 5-6 weeks in total.

2. Ceramide has many functions, among them the induction of cell death. However, it is too simple to describe ceramide as an exclusive pro-cell death molecular. In many systems it is a stress mediator, in some systems it even promotes cell growth and stimulation. Further, it is very important to discriminate the compartments in which ceramide has been generated and accordingly the enzymes or pathways that released or synthesized ceramide from pre-cursors. Several of the statements in the manuscript are too general.

Response: We apologize for general statements. These are now clarified throughout the text.

3. The data that link CerS1 to mitophagy are convincing and provide re-constitution experiments. They are done in one cell line only and need to be repeated in second cell line.

Response: To address this point, we have performed reconstitution studies in HNSCC cells, in which endogenous CerS1 was knocked down by shRNA, and effects of the expression of WT and mutant-CerS1 on mitophagy and growth inhibition in response to mitophagy inducer sodium selenite were measured. These data are now provided in Fig. 1D-E.

4. Fig. 2C shows a minimal difference between vector + cisplatin (red curve) and E7 and cisplatin (orange) only at doses between 6.25 and 25 nM cisplatin. The difference is biologically irrelevant. Further, E6-transfected cells also almost do not differ from the E7-transfected cells. At 50 nanomolar all cells are dead, at 25 nM approximately 30% vs 20% (E7 transfected). Such a small difference does not allow the very general conclusion in the present manuscript.

Response: We apologize for the confusing graph. We have now corrected the figure to clarify these data in Fig. 2C, which show changes in IC50 concentrations, which are significantly changed.

5. Knock-down of LC3B and Atg5 prevents C18-pyr-cer mediated mitophagy, but this knock-down will also prevent other forms of autophagy and does not prove that mitophagy is the mechanism by which E7 facilitates killing of the tumor cells.

Response: We agree, however, in our studies, including our previously published work (Sentelle et al, Nat Chem Biol, 2012, and Dany et al, Blood, 2016), data showed that C18-Pyr-Cer induces

mitophagy and not general autophagy, as Drp1-mediated mitochondrial fission is required for ceramide-mediated mitophagy. This is now mentioned in the text (p. 7).

6. It is difficult to understand how a dose dependent difference in mitophagy in the different cells treated with ceramide analogs fits to the complete absence of mitophagy in E7-negative cells.

Response: We apologize not to make this point clear. Ceramide-mediated mitophagy is detected in HPV- HNSCC cells at higher concentrations and/or time points (as reported in our previous study- Sentelle et al., 2012). This is now mentioned in the text (p. 9).

7. Several studies show trypan blue positive cells between 6 and 10% at maximum. This is again irrelevant in a tumor cell line.

Response: We apologize for unclear representation of these data. These figures for trypan blue studies are now clarified, as the differences are now reported as fold changes with significance values.

8. Fig. 1A, Fig. 2D (which is new 2E) and many other fluorescence microscopy images are overexposed and some of the co-localizations might be simply indicated by the overexposure. Please also quantify the overlap of the two signals.

Response: Data shown in 1A and 2E (old 2D) are now quantified and presented as suggested.

Referee #2 (Comments on Novelty/Model System):

We thank the reviewer for stating that “the authors are using a number of excellent techniques with high quality throughout the manuscript; no one has determined why HPV status correlates to chemotherapy outcomes, and this paper demonstrated why and orients the signaling pathway; model systems are standard and appropriate”.

Our point-by-point response to reviewer’s comments are as below:

Major criticism 1: All mitophagy panels: Although colocalization of lysotracker and mitotracker may be an indication of mitophagic flux, this method is not a definitive one, and is especially problematic as, by eye, the C18-ceramide treatment seems to enhance the mitotracker red intensity in some instances. It is also unclear as to how mitochondrial staining is enhanced with mitophagic signaling, as one would predict that targeting mitochondria for autophagy would reduce mitochondria number/function. The authors should provide some corroborating data such as mitochondria dyes for total numbers versus function and simple westerns for mitochondria.

Response: We agree. We have identified in our previously published manuscript that degradation of a mitochondrial matrix protein ACO2 (a mitochondrial matrix protein), provides a valuable marker for detection of mitophagy using Western blotting, in addition to co-localization of lysotracker and mitotracker. We now provide ACO2 degradation as an additional mitophagy marker in Figs. 1D, 2F and 10K. Degradation of mitochondria is also shown in Fig. 3C by ceramide-induced mitophagy.

Major criticism 2: Figures 3-4: There is some concern that the ceramide treatment used to induce mitophagy is having a general metabolic effect, especially as the appropriate controls have not been followed. Although later figures alleviate these concerns somewhat, using control species would enhance enthusiasm for this manuscript. Indeed, Pyr-Cer is in no way specific on many levels for either just mitochondrial targeting or chain length specificity when used as an exogenous agent.

Response: To address this point, we have performed additional studies using C18-Pyr-Cer, C16-Pyr-Cer, and C6-Pyr-Cer, which are targeted to mitochondria due to their cationic pyridinium conjugates, compared to conventional C18-ceramide, C16-ceramide and C6-ceramide (that are not targeted to accumulate in mitochondria). The data showed that only mitochondria-targeted long chain Pyr-C18-ceramide and C16-Pyr-Cer induced mitophagy. These data suggest that mitochondrial localization but not fatty acyl chain lengths of long chain ceramides regulate mitophagy induction. These data are now included in the revised manuscript (Fig. 3F).

Moderate criticism 1: Figure 2D-2E: The timing of cisplatin treatment and mitophagy is unclear in these figures. The authors claim that mitophagy (which is predicted to translate into fewer mitochondria/cell, as the mitochondria flux through the autophagic pathway) is occurring in Fig. 2D. However, in Fig. 2E (new 2F), the mitochondrial marker Tom20 is increasing with increased cisplatin treatment. Furthermore, the immunoblot quantification results seem not to match the image. Addressing these concerns would be valuable.

Response: We agree. We have repeated these studies and also included ACO2 as an additional mitochondrial marker in Fig. 2F.

Moderate criticism 2: Throughout the manuscript, the authors have not used appropriate statistical tests. Student's t-test is only appropriate when there are only 2 experimental conditions. An ANOVA with appropriate post-hoc test should be used in most instances in this manuscript.

Response: We apologize not to clarify the statistical details. The use of ANOVA is now mentioned in the text, especially for our in vivo studies.

Moderate criticism 3: Figure 9A-B and Figure 10H: The MFF/Drp1 colocalization micrographs are small and difficult to interpret. A coIP (similar to Figure 7E) is highly recommended to take their place.

Response: To address this point, we performed co-IP studies for MFF/Drp1 interaction to support studies shown in Fig. 9A-B (see new Fig 9C-D). Tumor tissue samples obtained from in vivo tumors shown in Fig. 10H did not contain sufficient amount of proteins to perform co-IP studies (in freshly frozen tissue sections).

Moderate criticism 4: Throughout the manuscript: Although the respiration data are very nice, some data correlating respiration rate to total mitochondria per cell would be more informative as to mitochondrial performance.

Response: To address this point, we have repeated a respiration study to measure the effects of C18-Pyr-Cer on oxygen consumption rate using the Sea Horse. These data are now included in the revised manuscript (Fig. 3A and EV2C). In addition, we have included ACO2 degradation as a marker for mitochondrial degradation in our studies as mentioned above (in response to major criticism 1).

Moderate criticism 5: Figure 3D: The data here demonstrate that ATG5 "knock-down" basally enhances mitochondria/lysosome colocalization. This is, at first glance, unexpected and should be addressed.

Response: We agree. The basal changes in mitochondria/lysosome co-localization in response to ATG5 or LC3 knockdown are not statistically significant.

Minor criticism 1: Throughout the manuscript, there are multiple mis-labeling errors. For example, in the legend for Figure 1 A-B mentions image quantification, but a graph does not appear in the figure. Furthermore, Figure 2 legend title "HPV-E6 enhances chemotherapy-mediated..." should likely read "HPV-E7 enhances...". In Figure 1D, the asterisks are askew and in Figure 6A, the line above the statistically significantly different shRNAs is not centered correctly.

Response: We apologize for these errors. These points are now corrected.

Minor criticism 2: Throughout the manuscript, panel labels are not centered and/or are small and thus difficult to read. In addition, some panels are labeled with the cell line used and others are not, and the labeling of many y-axes are inconsistent (for example, some read "trypan blue positive cells" and others read "fold trypan positive/total cells"). More consistent and larger labeling for clarity would be greatly appreciated. One should aspire to the Carmen Guidelines for Figure production especially for more high impact journals.

Response: We agree. We have now used more consistent labeling for graphs and figures throughout the manuscript.

Minor criticism 3: The authors make a few odd choices as to figure panel organization. For example, the wonderful pathway figure, which ties all the data together is relegated to the "expanded view" and not showcased in the main manuscript.

Response: The graphical abstract figure is now included in the main figures as Fig. 11.

Minor criticism 4: "Sea Horse" is not defined except in the methods section, which was confusing.

Response: The Sea Horse is now clarified in Materials and Methods, and also in the text.

Minor criticism 5: Figure 1A: Are the wild-type MEFs larger than the mutant cerS MEFs? By eye, based on the micrographs shown, it would seem so.

Response: These images are now adjusted to show equal sizes of cells in Fig. 1A.

Minor criticism 6: Figure 5A: Although the Rb "knock-down" shown originates on the same blot, a repeat run using sequential lanes might be useful.

Response: We agree. We now provide new data obtained from sequential lanes for these samples shown in Fig. 5A and EV3A (old EV2A).

2nd Editorial Decision

21 April 2017

Thank you for the submission of your revised manuscript to EMBO Molecular Medicine. We have now received the enclosed reports from the referees that were asked to re-assess it. As you will see the reviewers are now globally supportive and I am pleased to inform you that we will be able to accept your manuscript pending the following final amendments:

- 1) Thank you for providing a synopsis image. However, please note that it must be a 550 px-wide x 400-px high jpeg file.
- 2) The manuscript appears to be missing a callout for Fig. EV2A.
- 3) Certain panels in Fig 2D appear to be duplicated in Fig. EV2A. Please explain and add appropriate indications in the figure legend.
- 4) Image panels in Fig. 3B are very small and difficult to see. It would be best to increase their size if possible.
- 5) The scale bars in Fig. 10 panels J and I are barely visible. Please thicken and maybe change to colour to white.
- 6) Please indicate wherefrom the magnification insets are derived in the source images in Figs 3C, 4A, 4E, 5D, 6C, 8E and 10D
- 7) References with numerous authors are currently listed to show 10 authors followed by et al. Please correct them to show 20 authors et al.
- 8) As per our Author Guidelines, the description of all reported data that includes statistical testing must state the name of the statistical test used to generate error bars and P values, the number (n) of independent experiments underlying each data point (not replicate measures of one sample), and the actual P value for each test (not merely 'significant' or 'P < 0.05').
- 9) The manuscript must include a statement in the Materials and Methods identifying the institutional and/or licensing committee approving the experiments, including any relevant details (like how many animals were used, of which gender, at what age, which strains, if genetically

modified, on which background, housing details, etc). We encourage authors to follow the ARRIVE guidelines for reporting studies involving animals. Please see the EQUATOR website for details: <http://www.equator-network.org/reporting-guidelines/improving-bioscience-research-reporting-the-arrive-guidelines-for-reporting-animal-research/>. Please make sure that ALL the above details are reported.

10) We encourage the publication of source data, with the aim of making primary data more accessible and transparent to the reader. Would you be willing to provide a PDF file per figure that contains the original, uncropped and unprocessed scans of all or at least the key gels used in the manuscript and/or source data sets for relevant graphs? The files should be labeled with the appropriate figure/panel number, and in the case of gels, should have molecular weight markers; further annotation may be useful but is not essential. The files will be published online with the article as supplementary "Source Data" files. If you have any questions regarding this just contact me.

Please submit your revised manuscript within two weeks. I look forward to seeing a revised form of your manuscript as soon as possible.

***** Reviewer's comments *****

Referee #1 (Comments on Novelty/Model System):

This is an interesting manuscript showing a novel mechanisms of chemotherapy-resistance.

Referee #1 (Remarks):

The authors addressed all issues raised in my initial review.

Referee #2 (Comments on Novelty/Model System):

See previous review for these comments. This is a review of a revised manuscript.

Referee #2 (Remarks):

The authors made all requested revisions to the original manuscript, which are of good quality and support their conclusions.

2nd Revision - authors' response

01 May 2017

Thank you for your kind message and editorial points about our manuscript. Please find below our responses to the editorial points:

1) Thank you for providing a synopsis image. However, please note that it must be a 550 px-wide x 400-px high jpeg file.

Response: The synopsis image is now provided as a jpeg measuring 550 px wide by 400px high.

2) The manuscript appears to be missing a callout for Fig. EV2A.

Response: Figure 2EVA is now referred to in the text of the manuscript.

3) Certain panels in Fig 2D appear to be duplicated in Fig. EV2A. Please explain and add appropriate indications in the figure legend.

Response: The figure legend now explains that Fig. EV2A shows the extended time course for the experiment shown in Fig. 2D.

4) Image panels in Fig. 3B are very small and difficult to see. It would be best to increase their size if possible.

Response: The images in Fig. 3B have been enlarged and are now much easier to see.

5) The scale bars in Fig. 10 panels J and I are barely visible. Please thicken and maybe change to colour to white.

Response: The scale bars in Fig. 10 J and I have been enlarged and changed to white.

6) Please indicate wherefrom the magnification insets are derived in the source images in Figs 3C, 4A, 4E, 5D, 6C, 8E and 10D

Response: The areas from which the magnified portions came are now indicated in the above mentioned images.

7) References with numerous authors are currently listed to show 10 authors followed by et al. Please correct them to show 20 authors et al.

Response: All references have been corrected to show 20 authors et al.

8) As per our Author Guidelines, the description of all reported data that includes statistical testing must state the name of the statistical test used to generate error bars and P values, the number (n) of independent experiments underlying each data point (not replicate measures of one sample), and the actual P value for each test (not merely 'significant' or 'P < 0.05').

Response: Exact p-values, numbers of independent experiments (n), and statistical tests are now provided.

9) The manuscript must include a statement in the Materials and Methods identifying the institutional and/or licensing committee approving the experiments, including any relevant details (like how many animals were used, of which gender, at what age, which strains, if genetically modified, on which background, housing details, etc). We encourage authors to follow the ARRIVE guidelines for reporting studies involving animals. Please see the EQUATOR website for details: <http://www.equator-network.org/reporting-guidelines/improving-bioscience-research-reporting-the-arrive-guidelines-for-reporting-animal-research/>. Please make sure that ALL the above details are reported.

Response: The Materials and Methods section provides the institutional committee that approved the experiments, along with the number of mice, the genetic background of the animals, and that they were age- and sex-matched.

10) We encourage the publication of source data, with the aim of making primary data more accessible and transparent to the reader. Would you be willing to provide a PDF file per figure that contains the original, uncropped and unprocessed scans of all or at least the key gels used in the manuscript and/or source data sets for relevant graphs? The files should be labeled with the appropriate figure/panel number, and in the case of gels, should have molecular weight markers; further annotation may be useful but is not essential. The files will be published online with the article as supplementary "Source Data" files. If you have any questions regarding this just contact me.

Response: Source images are provided for Western blots as separate PDFs.

In addition, we have realized that two co-authors were missing in the previous version. Those authors are now included: Ryan De Palma and Jisun Kim.

We hope that this version is now acceptable for publication.

Corresponding Author Name: Besim Ogretmen

Manuscript Number: 2016-07088